# Particulate-Phase Mercury Emissions from Biomass Burning and Impact on Resulting Deposition: a Modelling Assessment

Francesco De Simone[1], Paulo Artaxo[5], Mariantonia Bencardino[1], Sergio Cinnirella[1], Francesco Carbone[1], Francesco D'Amore[1], Xin Bin Feng[6], Christian N. Gencarelli[1], Ian M. Hedgecock[1], Matthew Landis[7], Francesca Sprovieri[1], Noriuki Suzuki[3], Ingvar Wängberg[4], and Nicola Pirrone[2]

[1]CNR-Institute of Atmospheric Pollution Research, Division of Rende, UNICAL-Polifunzionale, 87036 Rende, Italy
[2]CNR-Institute of Atmospheric Pollution Research, Area della Ricerca di Roma 1, Via Salaria km 29,300, Monterotondo, 00015 Rome, Italy
[3]National Institute for Environmental Studies (NIES), Ministry of Environment, Okinawa, Japan
[4]IVL, Swedish Environmental Research Inst. Ltd., Göteborg, Sweden
[5]University of Sao Paulo, Sao Paulo, Brazil
[6]Institute of Geochemistry, State Key Laboratory of Environmental Geochemistry, Chinese Academy of Sciences, Guiyang, China
[7]Office of Research and Development, US Environmental Protection Agency, Research Triangle Park, NC, USA

*Correspondence to:* Francesco De Simone (f.desimone@iia.cnr.it)

**Abstract.** Mercury (Hg) emissions from Biomass Burning (BB) are an important source of atmospheric Hg and a major factor driving the inter-annual variation of Hg concentrations in the troposphere. The greatest fraction of Hg from BB is released in the form of elemental Hg ($Hg^0_{(g)}$). However, little is known about the fraction of Hg bound to particulate matter ($Hg^P$) released from BB, and the factors controlling this fraction are also uncertain. In light of the aims of the Minamata Convention

to reduce intentional Hg use and emissions from anthropogenic activities, the relative importance of Hg emissions from BB will have an increasing impact on Hg deposition fluxes. Hg speciation is one of the most important factors determining the redistribution of Hg in the atmosphere and the geographical distribution of Hg deposition. Using the latest version of the Global Fire Emissions Database (GFEDv4.1s) and the global Hg chemistry transport model, ECHMERIT, the impact of Hg speciation in BB emissions, and the factors which influence speciation, on Hg deposition have been investigated for the year

2013. The role of other uncertainties related to physical and chemical atmospheric processes involving Hg, and the influence of model parametrisations were also investigated, since their interactions with Hg speciation are complex. The comparison with atmospheric $Hg^P$ concentrations observed at two remote sites, Amsterdam Island (AMD) and Manaus (MAN) in the Amazon showed a significant improvement, when considering a fraction of $Hg^P$ from BB. The set of sensitivity runs also showed how the quantity and geographical distribution of $Hg^P$ emitted from BB has a limited impact on a global scale, although the

inclusion of increasing fractions $Hg^P$ does limit $Hg^0_{(g)}$ availability to the global atmospheric pool. This reduces the fraction of Hg from BB which deposits to the world's oceans from 71% to 62%. The impact locally is however significant, northern boreal and tropical forests where fires are frequent and uncontrolled leads to notable Hg inputs to local ecosystems. In the light of on-going climatic changes this effect could be potentially be exacerbated in the future.

# 1  Introduction

Emissions from biomass burning (BB) are an important source of mercury (Hg) to the atmosphere (De Simone et al., 2015; Friedli et al., 2009), and a major factor in determining the inter-annual variations of its tropospheric concentration (Slemr et al., 2016). Although the Hg released by BB varies from year to year, it can amount to up to roughly one third of the anthropogenic emission estimates (AMAP/UNEP, 2013; Friedli et al., 2009; De Simone et al., 2015). With the eventual implementation of the Minamata Convention (http://www.mercuryconvention.org/) and future curbs on industrial emission, as a by-product of industrial emission abatement measures, its relative importance will increase in the coming years. A previous modelling study (De Simone et al., 2015), used the global Hg chemistry model, ECHMERIT, and three BB inventories to assess the distribution of Hg deposition resulting from BB. A large part of the Hg released from BB deposits over oceans, where its re-emission is driven by sea surface temperature among other factors (Carbone et al., 2016; Andersson et al., 2011), or where it can be converted to toxic methyl mercury (MeHg) compounds, with important implications for the food web, and through fish consumption, also for human health (see Chen et al. (2016) and references therein). The deposition flux of Hg from BB has been shown to be more sensitive to certain factors, in particular the chemical mechanism employed in the model and the choice of emission inventory, than to others such as the vertical profiles of emissions (De Simone et al., 2015). In this previous study all Hg emitted from BB was considered to be $Hg^0(g)$. There is, however, evidence that the fraction of Hg emitted bound to particulates ($Hg^P$) may be sizeable, up to 30%, especially when the Fuel Moisture Content (FMC) is high (Obrist et al., 2007; Finley et al., 2009; Friedli et al., 2009; Wang et al., 2010). These levels however remain uncertain since different methodologies have led to different conclusions (Zhang et al., 2013; Obrist et al., 2007). Little is known about the mechanisms that control the speciation of Hg in BB emissions, which leads to uncertainties in the Hg deposition patterns, since the atmospheric lifetime of $Hg^P$, is significantly shorter than $Hg^0_{(g)}$, leading to greater local deposition. Local Hg deposition due to BB could have important repercussions in regions such as the South-East Asia where there is intensive rice cultivation, and which is subject to major BB events, especially during El Niño periods. Hg deposited to rice paddies can be readily converted to toxic MeHg that can accumulate in the grains (Wang et al., 2015; Feng et al., 2008; Meng et al., 2014; Zhang et al., 2010). Moreover it has been reported that $Hg^P$ from BB deposited to foliage has the ability to enhance MeHg formation (Witt et al., 2009). The aim of this study is to investigate the effects on simulated deposition fluxes of Hg resulting from BB, when variations in $Hg^P$ fraction and production processes are considered. The most recent version of the GFED BB emission inventory (van der Werf et al., 2010; Randerson et al., 2012; Mu et al., 2011), has been included in the global online Hg chemical transport model ECHMERIT, to simulate Hg deposition from BB for the year 2013 and to quantify the influence of variations in model inputs, assumptions and parametrisations.

## 2 Methods

### 2.1 The Biomass Burning Inventory

The reference BB inventory in this study, Global Fire Emissions Database version 4 (GFED4.1s), is based on an updated version of the inventory of van der Werf et al. (2010) with burned area from Giglio et al. (2013), and with the addition of small fire burned area (Randerson et al., 2012). The standard temporal resolution of the emissions files is monthly, however data are provided to distribute these daily, and a diurnal cycle based on Mu et al. (2011) is also available. Daily BB emissions from two other global inventories, GFASv1.2 (Kaiser et al., 2012, 2015) and FINNv1.5 (Wiedinmyer et al., 2011), were also included in the model for sensitivity runs. These three inventories are all compiled using the imagery obtained from the MODIS instruments. However, the way in which the data are filtered or processed yield substantial differences between the final products, see Andela et al. (2013) and references therein for a detailed description of the differences among the inventories.

### 2.2 Experimental Set-Up

The global Hg chemical transport model ECHMERIT (Jung et al., 2009; De Simone et al., 2014) uses T42 horizontal resolution (roughly 2.8° by 2.8° at the equator) and 19 vertical levels up to 10 hPa. Hg emissions from BB were included in the model by mapping them to CO emissions using the global averaged Enhancement Ratio (ER) of $1.96 \times 10^{-7}$ as obtained by Friedli et al. (2009) averaging field measurements from different biomes in various regions around the globe, including in plume measurements from the CARIBIC project (Ebinghaus et al., 2007). Previous modelling studies have used different ERs (De Simone et al., 2015; Holmes et al., 2010), however all these values were well within the range of uncertainty ($0.3 - 6.0 \times 10^{-7}$ , see Wang et al. (2015)). ECHMERIT, in the base configuration, includes the oxidation of $Hg_{(g)}^0$ to in $Hg_{(g/aq)}^{II}$ oxidation by $O_3$/OH in the gas and aqueous phases. OH and $O_3$ concentration fields were imported from MOZART (Model for Ozone and Related chemical Tracers) (Emmons et al., 2010). $Hg^P$ is assumed to be inert, whether it is emitted from anthropogenic activities or BB, and it is subject to transport and deposition processes but is not involved in any chemical reactions. Mechanisms and parametrisations used for calculating the dry and the wet deposition of the different Hg species are the same as described in Jung et al. (2009). Beyond this standard configuration a number of alternative processes and chemical mechanisms have been considered for this study, as explained in 2.3. Atmospheric reduction of $Hg_{(g/aq)}^{II}$ to $Hg_{(g)}^0$ has been included in many models to regulate the residence time of $Hg_{(g)}^0$ in the atmosphere. However, a number of the proposed mechanisms are unlikely to occur under most atmospheric conditions, or are based on empirical rates to better match the observations, see Kwon and Selin (2016) for a recent review. Due to this uncertainty, reduction was not included in this study. No further $Hg^P$ particulate matter (PM) dimension distributions other than the standard log-normal particle size distribution, as described in detail in (Jung et al., 2009), were considered in this study due to large uncertainties regarding the dynamic size range of PM emitted during BB, see Janhall et al. (2010) and references therein. GFED4.1s provides monthly burned area, fire carbon (C) and dry matter (DM) emissions (http://www.falw.vu/~gwerf/GFED/GFED4/). A script is provided to derive gaseous and PM emissions from DM fields making use of biome based emission factors based on Akagi et al. (2011) and van der Werf et al. (2010). The resulting

emission fields were then interpolated on to the ECHMERIT T42 grid using the mass conserving remapping function included in the Climate Data Operators (https://code.zmaw.de/projects/cdo).

## 2.3 Simulations and their Scope

The "Base" simulation used as the reference case in this study includes daily BB emissions from GFEDv4.1s, in which a global uniform fraction of $Hg^P$, equal to 15% of the total Hg emission is assumed. This value is within the range of observations (Obrist et al., 2007; Finley et al., 2009). However, since there are uncertainties in the proportion of $Hg^P$ emitted from BB (Zhang et al., 2013), further simulations were carried out with varying fractions of $Hg^P$ ( 0%, 4% and 30%). Simulations were also conducted mapping the 15% of the total Hg emitted as $Hg^P$ to the geographical distribution of different proxy chemical species (see Section 2.4). The shorter lifetime of $Hg^P$ with respect to $Hg^0_{(g)}$ potentially means that the vertical profile of the emissions could have an impact on the distribution of Hg deposition, as is the case for other speciated Hg emission sources (De Simone et al., 2016). Therefore two vertical profile parametrisations, and different emission injection time resolutions, were also included in the study. The principal vertical profile used (PBL-Profile) maps the Hg emissions uniformly within the Planetary Boundary Layer (PBL), whereas in the second, the vertical profile of the standard version of the ECHAM-HAM model was used (HAM-Profile) (Zhang et al., 2012). The HAM-Profile is equal to PBL-Profile when the PBL height is greater than 4000 $m$, otherwise, 75% of the emissions are placed within the PBL, and the remainder in the two layers above the PBL (17 and 8%). This threshold value is arbitrary, however it is the standard configuration of ECHAM6-HAM2 (Zhang et al., 2012; Veira et al., 2015). Biomass burning emissions from GFASv1.2 (Kaiser et al., 2012, 2015) and FINNv1.5 (Wiedinmyer et al., 2011), were also used in the study to assess uncertainty related to the satellite imagery processing and inventory compilation. These simulations primarily employ a O$_3$/OH $Hg^0_{(g)}$ oxidation mechanism. However, since the precise atmospheric Hg oxidation mechanism remains unclear (Hynes et al., 2009; Subir et al., 2011, 2012; Gustin and Jaffe, 2010; Gustin et al., 2015; Ariya et al., 2015), a number of runs were performed using a Br-based oxidation mechanism. Some studies (Steffen et al., 2014; Amos et al., 2012) suggest that the partitioning of reactive Hg species the between gas and particulate phases might be driven by air temperature and on the surface are of the aerosol present in the atmosphere. Therefore, two other simulations weer conducted including the temperature dependent gas particle partitioning described in Amos et al. (2012), one assuming BB Hg emissions to be only $Hg^0_{(g)}$, and another assuming a 15% of BB Hg emissions to be $Hg^P$. To estimate the ratio of Hg deposition from BB compared to anthropogenic sources, six further simulations were conducted including only anthropogenic emissions using the EDGAR (Muntean et al., 2014), AMAP2010 (AMAP/UNEP, 2013) and STREETS (Corbitt et al., 2011) inventories, employing the O$_3$/OH and Br oxidation mechanisms. This study covers a single year, 2013, chosen due to the availability of measurements from GMOS network (Sprovieri et al., 2016a, b; D'Amore et al., 2015). All simulations were performed for a full year, without the rapid re-emission mechanism (Selin et al., 2008), and were continued without further emissions for another 12 months to allow most of the 2013 Hg emissions to be deposited. Finally, a selection of simulations were re-run including Hg emissions from all sources,BB, anthropogenic emissions from AMAP2010 (AMAP/UNEP, 2013), dynamic ocean emissions, terrestrial emissions and re-emissions as described in De Simone et al. (2014), to evaluate model

performance against measurements and to evaluate the assumptions made in this study. Sources. A summary of the simulations performed can be found in Table 1.

## 2.4 BB Emission Speciation

The release of Hg from BB occurs prevalently as $Hg^0_{(g)}$. However, as mentioned previously a measurable fraction may be emitted as $Hg^P$ (Obrist et al., 2007; Friedli et al., 2009; Finley et al., 2009; Wang et al., 2010). No significant amounts of gaseous oxidised Hg ($Hg^{II}_{(g)}$) have so far been detected in BB emissions (Obrist et al. (2007) and references therein). The speciation of Hg emissions is of great importance, since it largely determines the atmospheric lifetime and hence the distance emitted Hg is transported in the atmosphere before deposition, as seen for other speciated Hg sources (Bieser et al., 2014). The fraction of $Hg^P$ released by BB determined in field and laboratory studies ranges from fractions of a few percent to over 30% (Obrist et al., 2007). The factors determining speciation, and whether $Hg^P$ is directly emitted or if it is the product of the oxidation of $Hg^0_{(g)}$ within the plume (Obrist et al., 2007; Webster et al., 2016) are not known. However, foliage, moisture content, fuel type, plant species and combustion proprieties certainly play a role. $Hg^P$ emissions were found to be well correlated with particulate matter (PM) and Organic Carbon (OC) emissions (Obrist et al., 2007). Obrist et al. (2007) found that $Hg^0_{(g)}$ is the dominant species in dry fuel combustion, whereas the fraction of $Hg^P$ becomes appreciable when FMC reaches roughly 30%, above which $Hg^P$ release appears to increase linearly with FMC. In the inventory used for the "Base" case both $Hg^0_{(g)}$ and $Hg^P$ follow the spatial distribution of CO emissions from BB, and 15% of the emitted Hg is considered to be $Hg^P$, see Figs. 1(a) and 2(a). Hg emission fields were also compiled in which the $Hg^P$ fraction of the total Hg emitted was mapped to OC and PM emissions, see Figs. 2(b and c). A further emission field was compiled in which the ratio of $Hg^0_{(g)}$ to $Hg^P$ is determined by the FMC, Figs. 1(b) and 2(d). A relationship was found to exist between $Hg^P$ emissions and the fire burn duration and severity, and combustion conditions (Obrist et al., 2007; Webster et al., 2016). In particular high $Hg^P$ fractions were observed during smouldering phases, whereas very low or undetectable $Hg^P$ levels were found during flaming combustion. These potential parametrisations were not investigated here due to the difficulty in finding a suitable proxy data set. Appendix A contains a more detailed description of the methods used to calculate the different Hg BB emission fields.

## 3 Results

### 3.1 Emissions

The total Hg emitted in 2013 based on the GFED inventory is roughly $400\,\text{Mg}$, at the lower end of the initial estimates ($675 \pm 240\,\text{Mg}$) (Friedli et al., 2009), but reasonable considering the natural variation of BB activity and the diminishing trend of the CO emission estimates in the latest inventory revisions (up to 50% for some years) (van der Werf et al., 2010). Considering 15% of the emissions to be $Hg^P$, in the BASE run this corresponds to approximately $340\,\text{Mg}\ Hg^0_{(g)}$ and $60\,\text{Mg}$ $Hg^P$. Interestingly the emissions of $Hg^P$ amount to $58\,\text{Mg}$ when relating the $Hg^P$ fraction to FMC. The exact amount of Hg emitted by BB in the different model runs is detailed in Table 1. The spatial distribution and the vertical profile of the emission

injection height, considering the PBL-profile for $Hg^0_{(g)}$ and $Hg^P$ in the different cases considered are shown in Figs. 1 and 2. Both the geographical and vertical distributions of the emissions of the Hg species reveal notable differences depending on the methodology used, particularly for $Hg^P$. Compared to the cases where $Hg^P$ emissions are mapped to CO and PM (Figs. 2(a-b) and (e-f)), mapping $Hg^P$ to OC and using the FMC to determine the speciation (Figs. 2(c-d) and (g-h)) result in enhanced $Hg^P$

emissions, above 60°N and over some areas the Amazon, Central Africa and East Asia as evident in Fig. 3. The timing and location of the enhanced $Hg^P$ emission at northerly latitudes could be particularly relevant for Hg deposition to the Arctic. From Fig. 3, it is evident how the geographical distribution of the $Hg^P$ to $Hg^0_g$ emission ratio differs with the assumptions considered. However for OC and FMC there is general agreement on the areas where the $Hg^P$ emissions are relatively higher especially in the North Hemisphere, and particularly for areas above 60°N. The agreement between OC and FMC is not surprising and is

related to the combustion characteristics that enhance OC emissions, i.e. lower combustion temperatures and the dominance of the smouldering phase of combustion Zhang et al. (2013), that are likely to occur where FMC is greatest.

## 3.2 Emission latitudinal profiles

The latitudinal profiles of $Hg^0_{(g)}$ and $Hg^P$ emissions, using the different approaches (Sect. 2.4) are shown in Figs. 4(a) and 4(b). For those emissions mapped to CO, only the 15:85 ($Hg^P$:$Hg^0_{(g)}$) speciation is reported for clarity. The differences in the

15 latitudinal profiles of the $Hg^0_{(g)}$ emissions (Fig. 4(a)) are sizeable only for the peaks north of 45°N, where the FMC based speciation has an $Hg^0_{(g)}$ fraction below 85%. The latitudinal profiles of $Hg^P$ emissions mapped to PM and CO look very similar over the entire domain (Fig. 4(b)), apart from a peak a few degrees north of the equator. The $Hg^P$ emissions mapped to OC and FMC differ from the PM and CO profiles, but are similar to each other between roughly 30°S and 60°N. South of 30°S $Hg^P$ emissions mapped to OC are higher, while peak $Hg^P$ emissions derived from FMC at 65°N (1.5 g km$^{-2}$ y$^{-1}$) are nearly 30%

greater than those derived from OC and roughly double those mapped to CO and PM. Moreover, in the FMC scenario the peak in $Hg^P$ emissions at 65°N are greater than the peak seen at 15°S (1.5 vs 1.4 g km$^{-2}$ y$^{-1}$). As is particularly evident in Fig. 4(c), the most notable differences among the different assumptions hypothesised, are above 60°N, where both the OC and the FMC cases agree on the location of the greatest $Hg^P$ emissions probably due to the linkage between OC emissions and combustion processes favoured by FMC (Zhang et al., 2013), and between 30°S and 45°S, where only OC and PM are greater than "Base".

A previous modelling study focusing on the fate of Hg from BB, where all emissions were considered as $Hg^0_{(g)}$, showed that the long atmospheric life of the elemental Hg smoothed the deposition latitudinal profiles compared to the emission profiles (De Simone et al., 2015). The four panels in Fig. 5 compare the normalised latitudinal deposition profiles obtained for the "Base" simulation with those obtained from the alternative $Hg^P$ emission scenarios by category. Fig. 5(a) demonstrates the very limited impact of the time resolution used for BB emissions, most likely due to the coarse horizontal resolution of the

model. The two vertical emission profiles (Fig. 5(b)) give deposition fields that are to all effects indistinguishable, even when considering varying temporal resolution of the BB emissions, whereas assuming all emissions to be in the first model level (with an average height of approximately 35 meters) leads to enhanced deposition near emission peaks. In this instance, the maximum deposition coincides with peak emission, at approximately 15°S, whereas in all other cases maximum deposition is shifted towards the equator.

The similarities in the latitudinal profiles of $Hg^P$ emissions when mapped to CO and PM are reflected in their deposition profiles (Fig. 5(c)). The relatively greater deposition north of $60°N$ seen in Fig. 5(c) obtained when $Hg^P$ emissions are mapped to OC and when driven by FMC, reflect the peak in $Hg^P$ emissions at this latitude. The greatest differences in the latitudinal deposition profiles, using the GFED inventory, are seen when varying the percentage of $Hg^P$ in the emissions (Fig. 5(d)). Considering emissions to be solely $Hg^0_{(g)}$ yields a relatively smooth profile extending from pole to pole, increasing $Hg^P$ causes enhanced deposition near BB hot-spots. The emission peak at around $50°N$ remains relatively distinct also in the deposition for all the simulations (although it seen as a shoulder in the 100% $Hg^0_{(g)}$ profile). The peak north of $60°N$ is more dependent on emission speciation, supporting the previous finding that the location of Hg deposition depends on complex interactions between emission location and the time of year which influences both atmospheric transport patterns and oxidant concentration fields (De Simone et al., 2015).

## 3.3    Geographical Distribution of Hg Deposition

Due to the uncertainty in the atmospheric oxidation pathway of Hg, simulations were performed using both $O_3$/OH and Br oxidation mechanisms to investigate their impact on Hg deposition fields. Figure 6(a-d) compares the geographical distribution of the modelled Hg deposition field using emission fields with 0% and of 15% $Hg^P$, for each of the oxidation mechanisms. The $O_3$/OH mechanism leads to enhanced deposition in the tropics, whereas the Br mechanism leads to relatively higher deposition over the South Atlantic and Indian oceans. Assuming a fraction of $Hg^P$ in the emissions subtracts some $Hg^0_{(g)}$ from the global pool, and this fraction is deposited nearer to emission sources in Central Africa, South-East Asia, the Amazon, and near the wildfires which occur in North America and in North Asia in the northern hemisphere summer. From Fig.6, it appears that assuming a fraction of the BB emissions to be $Hg^P$ causes the deposition field simulated using the Br oxidation mechanism to more closely resemble that using the $O_3$/OH mechanism. To better understand the combined effect of Hg speciation and oxidation pathway on the deposition distribution, agreement maps were created, to highlight the similarities and differences in the distribution of high-deposition ($\geqslant \mu + 1\sigma$, the average plus 1 standard deviation) model cells in the different simulations as described in (De Simone et al., 2014). Figs. 7(a) and 7(b) show the agreement maps of the deposition for three different $Hg^P$ fractions using the two oxidation mechanisms. Using the $O_3$/OH mechanism, the number of model cells in which the model predicts high deposition in all three emission speciation scenarios is higher than when using the Br mechanism (631 *vs.* 248). This is due to the combination of high emissions and high oxidant concentrations in the Tropics when using the $O_3$/OH mechanism, constraining Hg deposition to a relatively narrow latitude band. Using the Br mechanism, Hg has a greater possibility of being transported to mid and high latitudes before being oxidised and deposited. In both the oxidation scenarios the higher deposition over the remote areas of North America and North Asia occurs only when the fraction of $Hg^P$ in the emissions is greater than zero. High local contributions to Hg deposition from BB using the Br mechanism occur more frequently when the fraction of $Hg^P$ is non-zero, purple in Fig. 7(b), unlike the $O_3$/OH simulations. Figure 8 contrasts the results from the two oxidation mechanisms with varying percentages of $Hg^P$, and a simulation in which the $Hg^P$ fraction was assumed to be 100%, so that it behaves as an inert tracer. The agreement maps show clearly that the similarity in the deposition fields increases with

increasing $Hg^P$ fraction, reflected in the number of cells where all three simulations agree (grey in the figure) and the decrease in the number of cells where only one simulation predicts deposition higher than $\mu + \sigma$, (red, blue and yellow).

## 3.4 Constraints from Global Measurements networks

The output from the simulations including all emissions (as indicated in Table 1) for the year 2013 were compared to measurement data available from GMOS and other monitoring networks. The sites are the same as those used in Travnikov et al. (2016), the measurements from which have been reviewed Sprovieri et al. (2016a) and Sprovieri et al. (2016b). Table 6 summarises a selection of metrics from the comparison for Total Gaseous Mercury ($Hg^0_{(g)} + Hg^0_{(g)}$) and for Hg in wet deposition. The results are in line with those obtained from previous studies (De Simone et al., 2015, 2016) focusing on a different time period, and indicate a generally good agreement between measured and simulated TGM, especially for the run with the Br driven oxidation mechanism. For the Hg wet deposition fluxes, the results show poorer performance, due to the difficulties for coarse resolution global models to simulate precipitation events correctly (De Simone et al., 2014; Roeckner et al., 2003). Since the different sensitivity runs considering $Hg^P$ from BB differ by a only a small perturbation in the speciation of total Hg emitted from the BASE (or the relevant reference) case, the results are actually indistinguishable from BASE (or the relevant reference) case. Therefore the table reports the comparison only from runs which yield different results. Also, this means that neither wet deposition nor TGM are the most appropriate variables to assess the validity of any of the assumptions concerning $Hg^P$ emitted during BB. During 2013, within the GMOS and other Hg monitoring initiatives, a number of measurement sites collected samples of atmospheric $Hg^P$. These stations and their precise locations are reported in the Table 2. The result of the comparison with the measurements from these sites is summarised in Fig. 9. Panel 9(a) shows the annually averaged surface concentrations of $Hg^P$ as simulated by the BASE run for 2013. As is evident, surface $Hg^P$ hots spot are close to the industrial areas of Eastern Europe, India, East Asia and South Africa, and to areas characterised by significant BB activity, including Indonesia, Central Africa and boreal areas of Canada and Asia.

A first analysis to find those areas where the model run assuming a fraction $Hg^P$ from BB (i.e. BASE) gives results that are statistically distinguishable from the model run assuming Hg from BB to be only $Hg^0_{(g)}$, was performed to identify the measurements sites best suited for further analysis.

The geographical distribution of these differences is reported in panel(b) of Fig. 9. The areas were the anthropogenic input is the greatest differ little between the simulations (based on a student t-test at 95% level of confidence), as indicated by dot points in the panel. Most of the stations, depicted by the blue solid points in the same panel, are within these regions, and therefore unsuitable for the analysis. Only three stations are in areas where the model results are significantly different. These, short names of which are reported in the panel, are Amsterdam Island (AMD), Manaus (MAN) and Muana Loa (MAU). However MAU and Mt. Waliguan (MWA) are high altitude sites and affected by processes other than BB. For both the remaining stations (AMD and MAN), the fraction of $Hg^P$ that is assumed to be emitted by anthropogenic activities, as estimated by AMAP2010 inventory (AMAP/UNEP, 2013), is not sufficient alone to explain the averaged $Hg^P$ concentrations collected over the year, as is evident from Fig. 9(c). The inclusion of 30% $Hg^P$ from BB emissions at MAN and AMD, and also the inclusion of 15% $Hg^P$ from BB as using the FINN inventory at MAN, significantly improve the model performances, in terms of the annual

average $Hg^P$ concentrations. The result of the comparison between the $Hg^P$ concentrations collected at these two stations with the same modelled at the same points by a selection of sensitivity runs at an finer temporal resolution (daily averages) is reported in the two panels of Fig. 10. The same comparisons for all the stations, among with the box and whisker plot of distributions of the $Hg^P$ concentrations measured and modelled, are reported in Fig. 11. Although the measurement coverage of the year at MAN is sporadic, it is an important station because it is situated in a remote area where the local Hg emissions are due only to ASGM (only $Hg^0_{(g)}$) and BB (Sprovieri et al., 2016b). The consistent reduction of the error between measured and modelled $Hg^P$ concentrations when consider a fraction of particulate bound Hg emitted from BB (NRMSE from 48% to 34% and 27% for 30% $Hg^P$ and FINN, respectively) clearly indicate the role of BB on the observed $Hg^P$ values. At AMD (Fig. 10 (b)), the inclusion of the fraction of $Hg^P$ from BB results only in a slightly better agreement with the measurements (NRMSE from 16% to 14%). However, the $Hg^P$ event matching grows from 25% to 32%, especially in the last part of the year, that a previous study have been associated with BB events in the central Africa (Angot et al., 2014). Peaks was evaluated using the "findpeak" function in MATLAB, available from https://it.mathworks.com/help/signal/ref/findpeaks.html. To summarise, it seems that the emissions of a fraction $Hg^P$ from BB is plausible and supported by the measures of atmospheric $Hg^P$, at least for the period investigated and for the location of the two remote stations AMD and MAN. However, it has to be noted that the uncertainties related to the precise nature of atmospheric $Hg^P$ and to the processes it undergoes in the atmosphere could have an appreciable impact on the model results. For example, the assumption of a temperature dependent gas-particle $Hg^{II}$ partitioning proposed by Amos et al. (2012) (i.e. the "Partitioning" and "Partitioning ref" runs) yield overall better model agreement with annually average $Hg^P$ concentrations (stars in Fig. 9(c)). However, comparing the modelled daily average time series with measurements results in clearly poorer performance at both the AMD and MAN stations, see Fig. 12(b) and (c). More importantly, this assumption tends to render statistically indistinguishable (student t-test at 95% level of confidence) the contribution of any eventual $Hg^P$ from BB, as evident from Fig. 12(a).

### 3.5 Uncertainty and Biomass Burning *versus* Anthropogenic Impact

Besides the uncertainty related to the atmospheric Hg oxidation mechanism (Hynes et al., 2009; Subir et al., 2011, 2012; Gustin et al., 2015; Ariya et al., 2015) there are a number of other factors that lead to uncertainty in ascertaining the fate of Hg released by BB. Some of the model assumptions and parametrisations, in particular emission height made little difference to the eventual deposition fields in the case where emissions from BB were considered to be 100% $Hg^0_{(g)}$ (De Simone et al., 2015). Other sensitivity studies of the speciation of anthropogenic emissions reveals that varying the fractions of $Hg^{II}_{(g)}$ and $Hg^P$ can result in quite different Hg deposition patterns, due to their shorter residence time compared to $Hg^0_{(g)}$ (De Simone et al., 2016; Bieser et al., 2014).

However the choice of the two main vertical profile of the BB emissions used in this study, also when combined with the temporal resolution of the emissions actually have little influence on the final Hg deposition fields. Emitting all, or part, of the Hg in to a single model layer does have an impact. However these cases are a little speculative, and therefore not included in the final analysis. The factor which has the greatest influence on the Hg deposition pattern is the choice of emission inventory, whereas for a given inventory the most important factors are the fraction of $Hg^P$ and the oxidation mechanism, although as

seen in Sect. 3.3 the impact of the oxidation mechanism decreases with increasing $Hg^P$ fraction. The method of calculating the $Hg^P$ fraction has a limited impact on deposition on a global scale, with 66% of Hg deposited over the oceans, but the regional impact does change. Using FMC to determine the $Hg^P$ fraction increases deposition to the Arctic by 16 and 13% ($O_3$/OH and Br, and to the Southern Ocean by 30 and 25% ($O_3$/OH and Br), see Table 4. Apart from the Polar oceans the oceanic basins most influenced by the fraction of $Hg^P$ in the BB emissions are the North and South Pacific and the Indian ocean. The total deposition to individual basins from the limiting 0 and 30% $Hg^P$ cases are included in Table 4. The horizontal pattern correlation method (Santer et al., 1995, 1996) and the non-parametric Kolmogorov-Smirnov two-sample test were used to assess the differences in the deposition fields obtained from the simulations summarized in Table 1, as in De Simone et al. (2015). The results of the comparison of the simulations with the "Base" run are presented in Table 3. The results of the Kolmogorov-Smirnov two-sample test were exploited to construct an inspected ensemble, following the approach of Solazzo and Galmarini (2015), and previously employed in De Simone et al. (2015). The ensemble includes only those simulations with realistic assumptions and deposition fields with little or no probability of belonging to the same distribution. Hg deposition from the resulting ensemble is shown in Fig. 13(a). The figure shows how the inclusion of $Hg^P$ in the BB emissions causes greater deposition near the hot spots of central Africa, Brazil, South-East Asia, North America and North Asia. Nonetheless approximately 70% of Hg deposition occurs over the oceans, with the Tropical Atlantic, Tropical Pacific and Indian Oceans most impacted (see Table 5). Figure 13(b) compares the BB ensemble results with an ensemble constructed using only anthropogenic emissions, using the EDGAR (Muntean et al., 2014), AMAP2010 (AMAP/UNEP, 2013) and STREETS (Corbitt et al., 2011) inventories, (considering both oxidation mechanisms (see Table 1). It can be seen that the contribution of BB to Hg deposition is close to or greater than that from anthropogenic activities in the areas near the locations of wildfires, central Africa, the Amazon, part of the Southern Atlantic and North Asia. The contribution to Hg deposition from BB relative to anthropogenic emissions is greater than 25% everywhere in the Southern Hemisphere, and exceeds 30% in the South Pacific and South Atlantic, table 5. As anthropogenic Hg emissions decline the relative impact of BB Hg will rise, as shown in Fig. 14, where the Hg deposition due to BB is compared with Hg deposition from anthropogenic sources in three different emission scenarios for 2035, see Pacyna et al. (2016) for details of the emission scenarios.

## 4   Conclusions

That a fraction of $Hg^P$ is present in BB Hg emissions has been confirmed by several field measurements (Obrist et al., 2007; Finley et al., 2009), and this fact has bee suggested as an explanation of high $Hg^P$ observations at a remote site (Angot et al., 2014), but this is the first time it has been included in a model study to assess its effects on a global scale.

A previous modelling study assuming emissions from BB to be 100% $Hg_{(g)}^0$ (De Simone et al., 2015) suggested that as much as 75% of the Hg emitted by BB was deposited to ocean basins, with global implications for food webs and human health. Including a fraction of $Hg^P$ in the BB Hg emissions has an impact on the geographical distribution of the deposition fluxes for the year analysed, reducing input to the global oceans and some high latitude regions, while enhancing potentially negative effects on ecosystems close to areas where significant BB occurs. The presence of $Hg^P$ in the emissions decreases

the differences seen in Hg deposition patterns produced by employing different oxidation mechanisms. In the remote areas of North Asia and North America, BB has a strong local impact if the $Hg^P$ fraction is non-zero. This latter result is independent of the atmospheric oxidation pathway. In simulations with 30% $Hg^P$ in the BB emissions, deposition over the Arctic increases by 11% with respect to 0% $Hg^P$ (30% in the Br simulations), and by 16% when the $Hg^P$ fraction is determined by FMC (37%

in the Br simulation). The fraction of $Hg^P$ released from BB while having an impact on the land-sea distribution of global Hg deposition, has a more significant impact in particular regions including the Polar regions, the South Atlantic and Pacific and Indian Oceans. These results apply for the investigated year (2013) and may differ for other years, due to the complex interaction of the numerous factors determining the final fate of Hg. However few alternatives of analysis period exist due the limited time coverage of global measurement network(s). Indeed the year selected for the analysis, allowed for the hypotheses

tested in this study to be supported by observations at a number of sites from GMOS which has extended the observational network in the Tropics and the Southern Hemisphere (Sprovieri et al., 2016a, b). The eventual emissions of a fraction of $Hg^P$ from BB cannot be evaluated by comparison with observed atmospheric Hg concentrations or Hg in wet precipitation samples, due to the very small impact of $Hg^P$ from BB on both the atmospheric burden and wet deposition relative to all other emissions sources ($\approx$ 1-2% ). Conversely, its contribution to atmospheric $Hg^P$ is comparable to that of anthropogenic

activities and therefore may be investigated. The inclusion in the model run of a fraction of $Hg^P$ from BB contributes to better model performances at two remote sites, Manaus, and Amsterdam Island. However results are not definitive, due to the large uncertainty related to $Hg^P$ emissions and transformation processes. Further modelling, and more measurement sites, particularly in remote areas, would help reduce some of the uncertainties associated with Hg emissions from BB, and to constrain these processes. Biomass burning has and will continue to play a significant role in the cycling of legacy Hg, and its

relative importance is likely to increase as anthropogenic emissions are reduced and global temperatures rise.

## Appendix A:  How Hg emission fields are calculated

### A1  Mapping to CO

When mapped to CO, the emissions of $Hg_g^0$ were calculated from those of CO using a global averaged ER ($1.96 \times 10^{-7}$). These were unchanged in the run assuming Hg emissions from BB to be 100% $Hg_g^0$, and divided between $Hg^P$ and $Hg_g^0$ species, with

ratios $4:96$, $15:85$, and $30:70$, in mass, in the runs considering the respective constant fractions of $Hg^P$. Consequently, the geographical and temporal distributions of $Hg_g^0$ and $Hg^P$ BB emissions follow those of CO. For all cases, the GFEDv4 inventory was used based, except for those sensitivity runs performed to test the impact of different inventories, FINNv1.5 and GFAS1.4.

### A2  Mapping to OC

When mapped to OC, geographical and temporal distributions of $Hg_g^0$ BB emissions, as well as the total Hg emitted, were calculated as described in Appendix A1. The fractioning of Hg emissions, in mass, between $Hg^P$ and $Hg_g^0$ species were

assumed to be in the ratio $15 : 85$. The Hg$^P$ emissions so calculated were then geographically and temporally mapped to those of OC from the GFEDv4 inventory.

## A3 Mapping to PM

This mapping method is similar to the one described in Appendix A2, except for the fact the Hg$^P$ temporal and geographical
distributions follow those of PM from the GFEDv4 inventory.

## A4 Emissions speciation determination by FMC

When using this procedure for determining the BB emission speciation between Hg$_g^0$ and Hg$^P$, the geographical and temporal distributions of Hg$_g^0$ and Hg$^P$ BB emissions, as well as the total Hg emitted, were calculated in the same way as described in Appendix A1. The main difference is in that the fractioning of Hg emissions, in mass, between Hg$_g^0$ and Hg$^P$ species were cal-
culated dynamically using the piece wise linear relationship between Fuel Moisture Content empirically determined by relative figure in Obrist et al. (2007). As a proxy for FMC, we used the monthly averaged vegetation water content (VWC) derived from passive microwave remote sensing data (Advanced Microwave Scanning Radiometer 2 (ASMR2)), and employing the Land Parameter Retrieval Model (LPRM) available at (http://gcmd.nasa.gov/search/Metadata.do?Entry=C1235316240-GES_ DISC#metadata).

*Acknowledgements.* We are grateful to Sebastian Rast and colleagues at the Max Planck Institute for Meteorology in Hamburg, Germany for the distribution of their software ECHAM5 and for providing the access to the processed ERA-INTERIM data. We are grateful to Xin Yang for providing the Br/BrO fields from p-Tomcat. The research was performed in the framework of the EU project GMOS (FP7–265113). The authors would also like to thank the referees whose helpful suggestions and comments contributed much to improving the original manuscript.

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

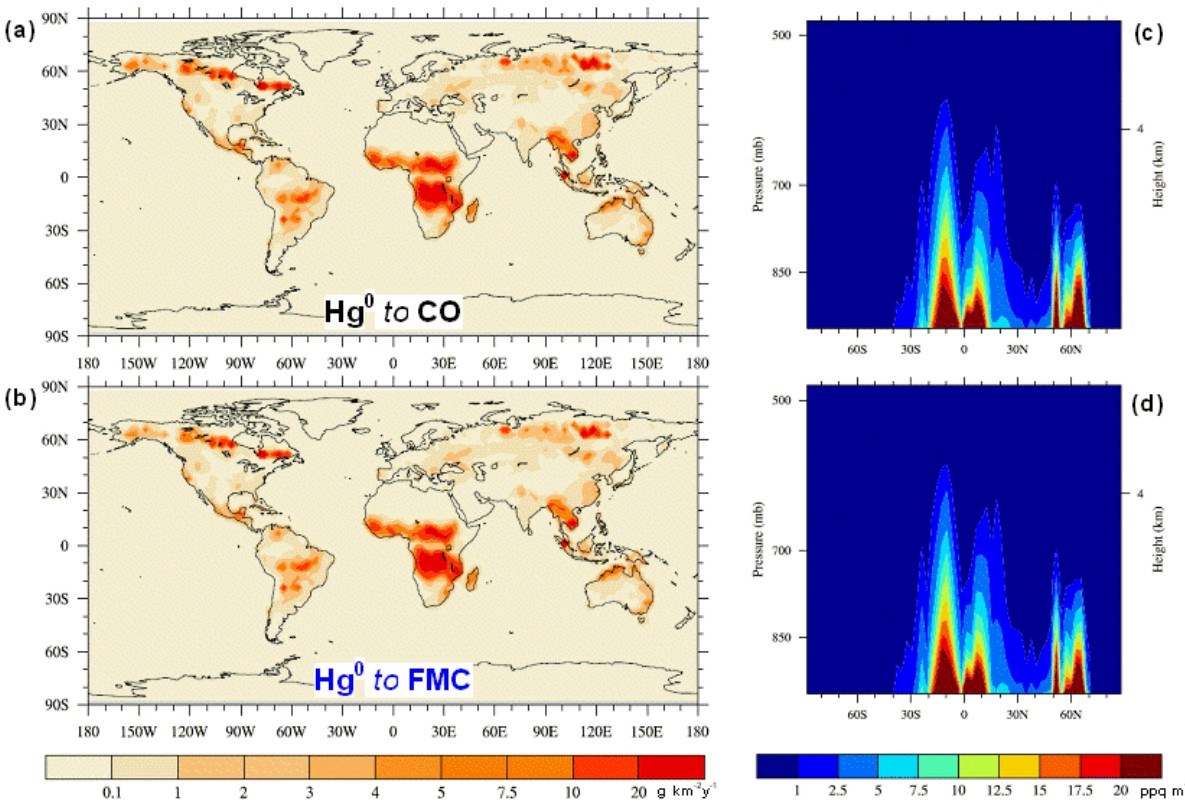

**Figure 1.** Geographical distribution (a-b)) and PBL-type vertical profiles (c-d) of the $Hg^0_{(g)}$ emissions, when mapped to CO (a,c) and when speciation is determined by FMC (b,d). For the emissions mapped to CO, only the speciation (15:85 $Hg^P$:$Hg^0_{(g)}$) is shown for clarity.

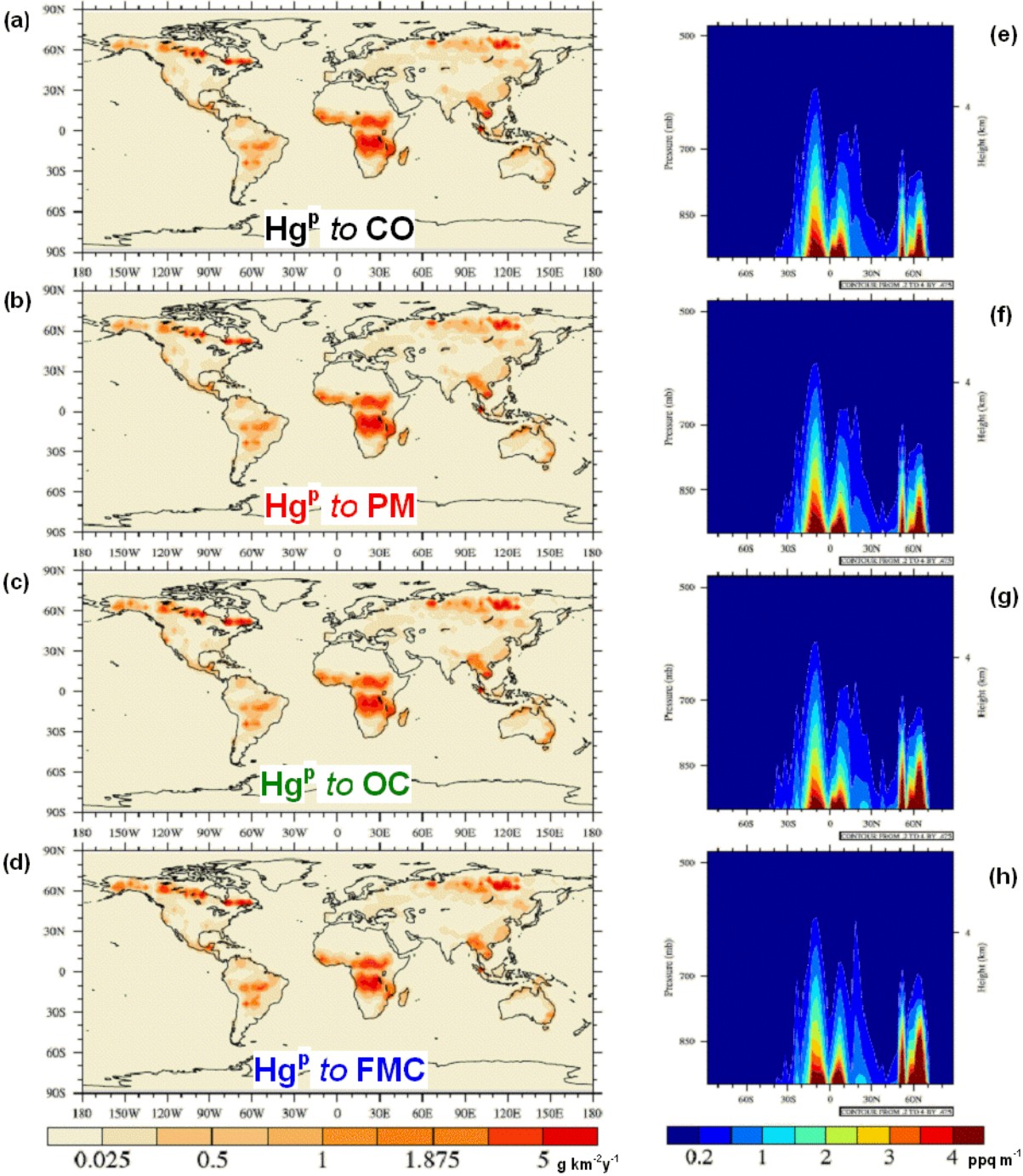

**Figure 2.** Geographical distribution (a-d)) and PBL-type vertical profiles (e-h) of the $Hg^P$ emissions as injected in the model, when mapped to CO (a,e), PM(b,f) and OC(c,g), and when speciation is determined by FMC (d,h). For the emissions mapped to CO, only the speciation (15:85 $Hg^P$:$Hg^0_{(g)}$) is shown for clarity.

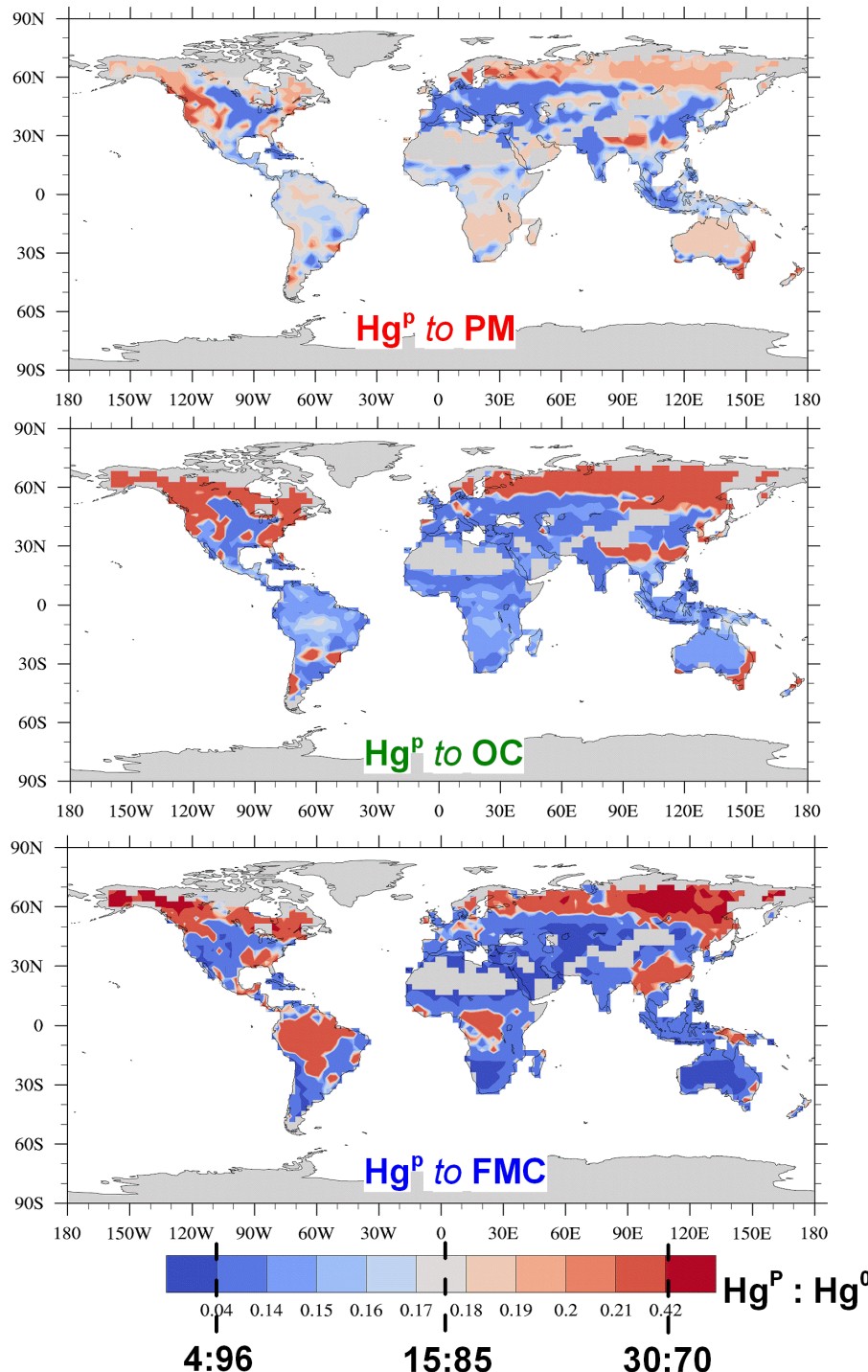

**Figure 3.** Geographical distribution of the $Hg^P : Hg^0_{(g)}$ emissions ratio, when mapped to PM(a) and OC(b), and when speciation is determined by FMC (c). In the color bar are indicated the levels corresponding to the constant speciations (4:96, 15:85 and 30:70 $Hg^P:Hg^0_{(g)}$) used for the emissions mapped to CO.

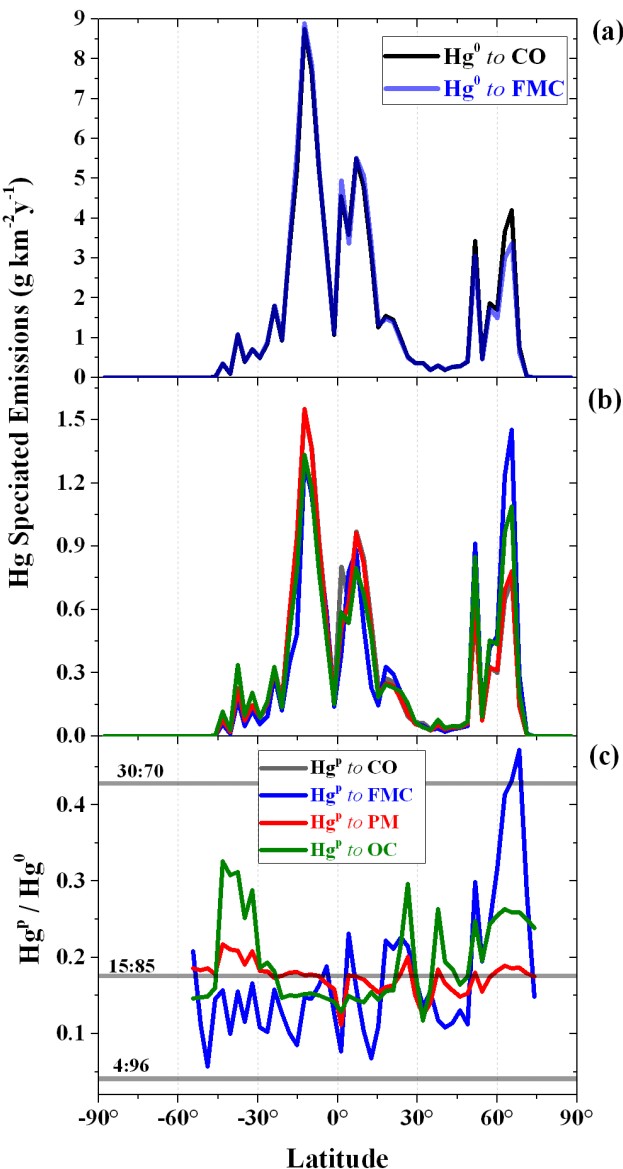

**Figure 4.** Latitudinal profiles of (a) $Hg_{(g)}^0$ emissions when mapped to CO and when speciation is determined by FMC; (b) $Hg^P$ emissions when mapped to CO, PM, OC, and when speciation is driven by FMC, respectively and of (c) the relevant ratio $Hg^P : Hg_{(g)}^0$ . For both $Hg_{(g)}^0$ and $Hg^P$ emissions mapped to CO, only the speciation (15:85 $Hg^P:Hg_{(g)}^0$) is reported for clarity , whereas in panel (c) all the speciations are reported

.

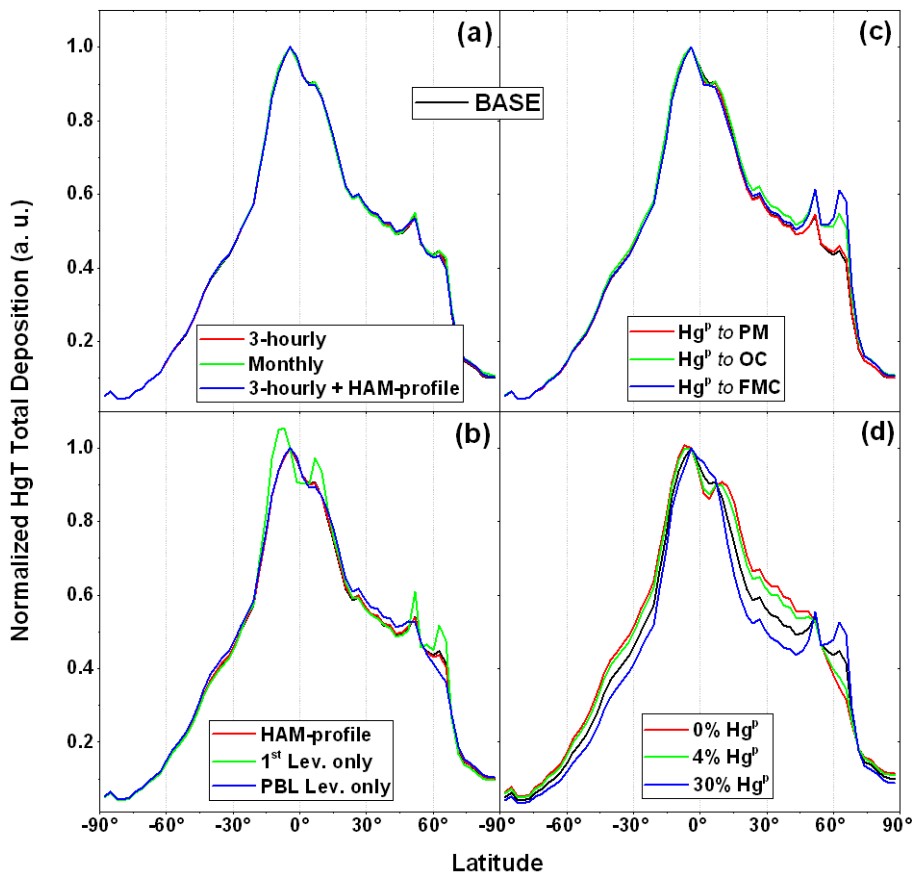

**Figure 5.** Latitudinal profiles of the normalised Hg total deposition from the model "Base" run, compared with a selection of sensitivity runs, assuming: (a-b) different emission time resolution and vertical profile, and a combination of both, (c) different $Hg^P$ emission geographical distributions, and different $Hg^0_{(g)}$:$Hg^P$ ratios. The normalisation was done by maximum.

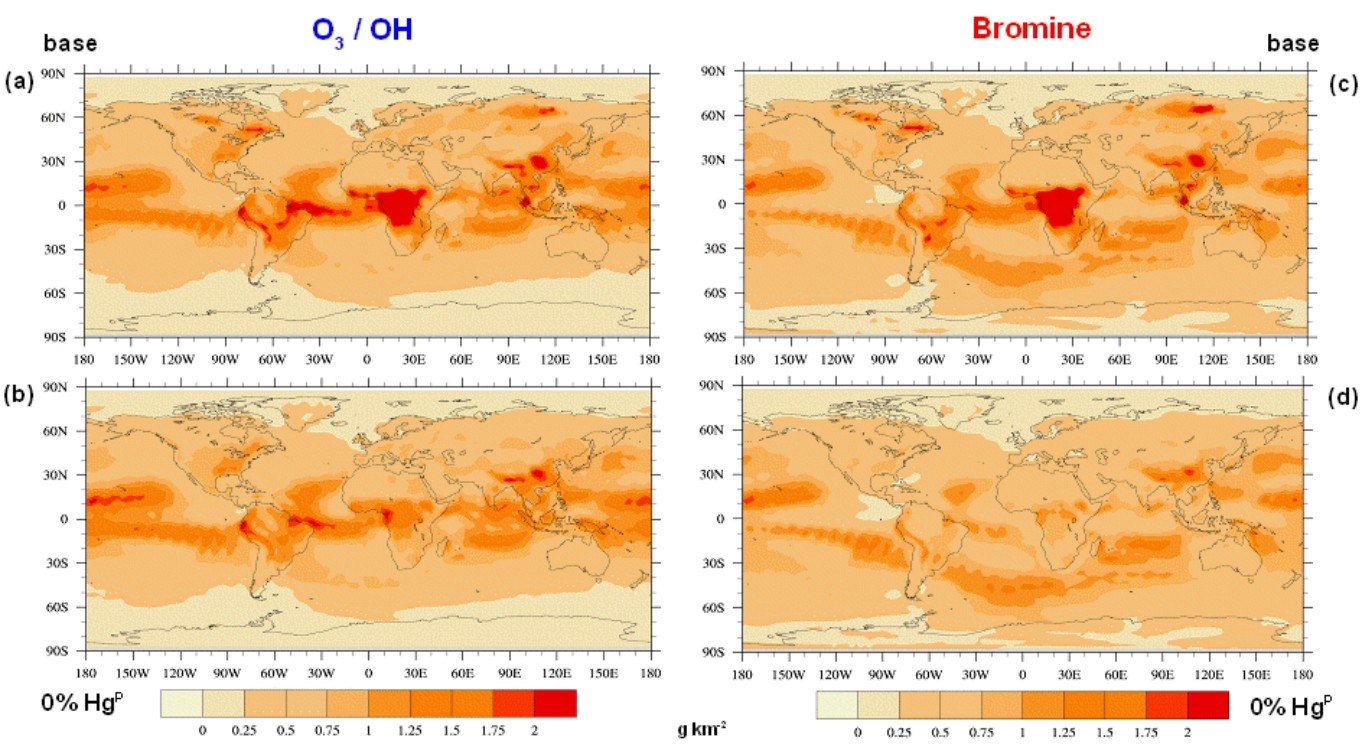

**Figure 6.** Geographical distribution of the Hg total deposition from model runs including only BB emission sources and assuming two different Hg$^P$ emission fractions, 15% (a,c) and 0% (b,d), for the two oxidation mechanisms considered, O$_3$/OH (a-b) and Br (c-d).

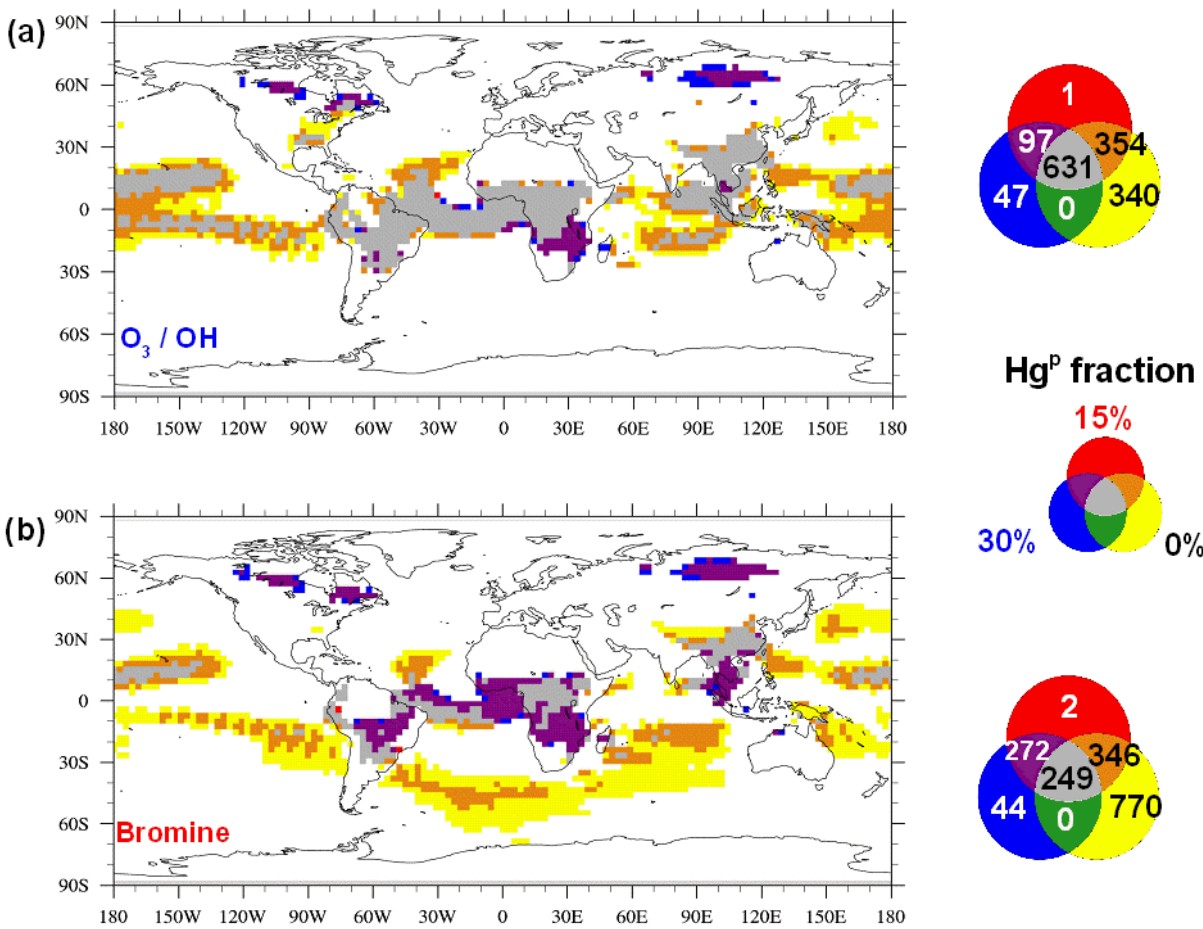

**Figure 7.** Agreement maps of high Hg deposition model cells obtained considering only BB emissions and assuming 0%, 15% and 30% to be Hg$^P$ under both the oxidation mechanisms considered, O$_3$/OH (a) and Br (b). The maps show the areas where deposition is greater than $\mu + \sigma$.

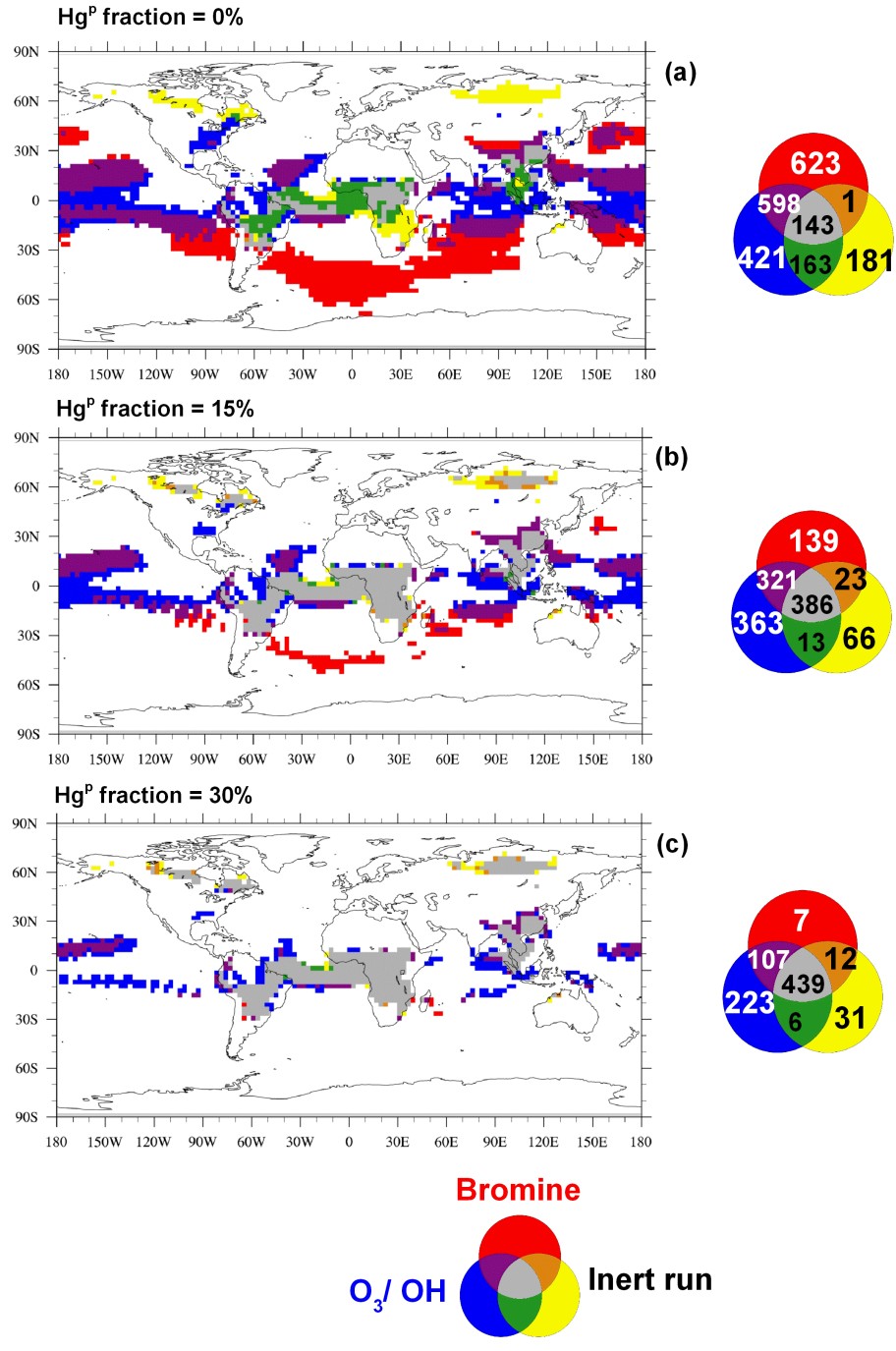

**Figure 8.** Agreement maps, under three different speciation scenarios: 0% (a), 15% (b), and 30% (c) $Hg^P$, of high Hg deposition model cells obtained considering only BB and using the $O_3$/OH, and the Br oxidation mechanisms, and a sensitivity run where all Hg BB emissions were considered inert (i.e. all $Hg^P$). The deposition field from for this "inert" run was retained under the three different speciation scenarios. The maps show the areas where deposition is greater than $\mu + \sigma$.

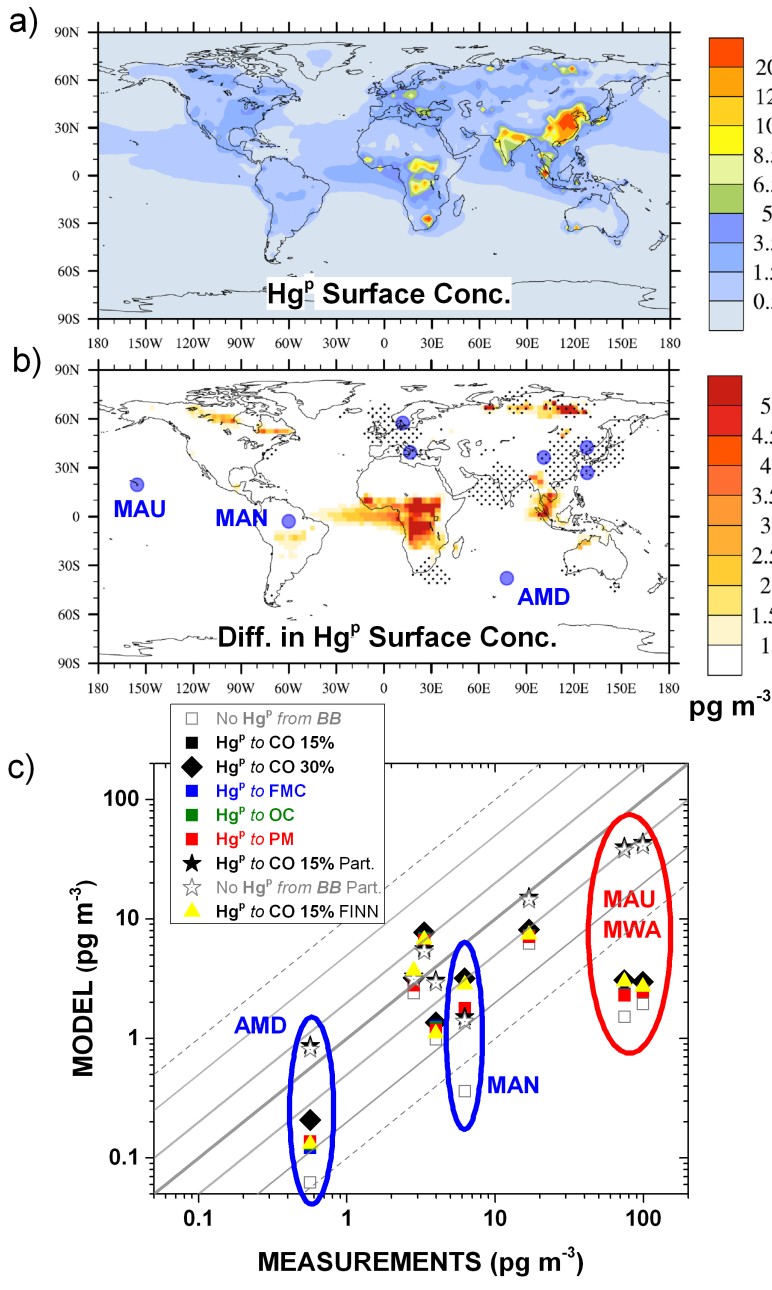

**Figure 9.** (a) Annual averaged surface Hg[P] concentrations as simulated by BASE run including all emission sources. (b) Differences in annual averaged surface Hg[P] concentrations as simulated by BASE and by NO Hg[P] runs, both including emissions from all sources. Black dots indicate that differences are not significant based on a student-t-test at a 95 % confidence interval. Blue bigger points indicate the locations of measurements sites reported in Table 2. Short names are depicted for sites where the differences between BASE and NO Hg[P] runs are significant. (c) Scatter plot of annual averaged Hg[P] concentrations measured at sites of Table 2, compared with those obtained by different sensitivity runs. The circles in the figure indicate values relative to the sites further investigated at an higher temporal resolution, see Fig. 10.

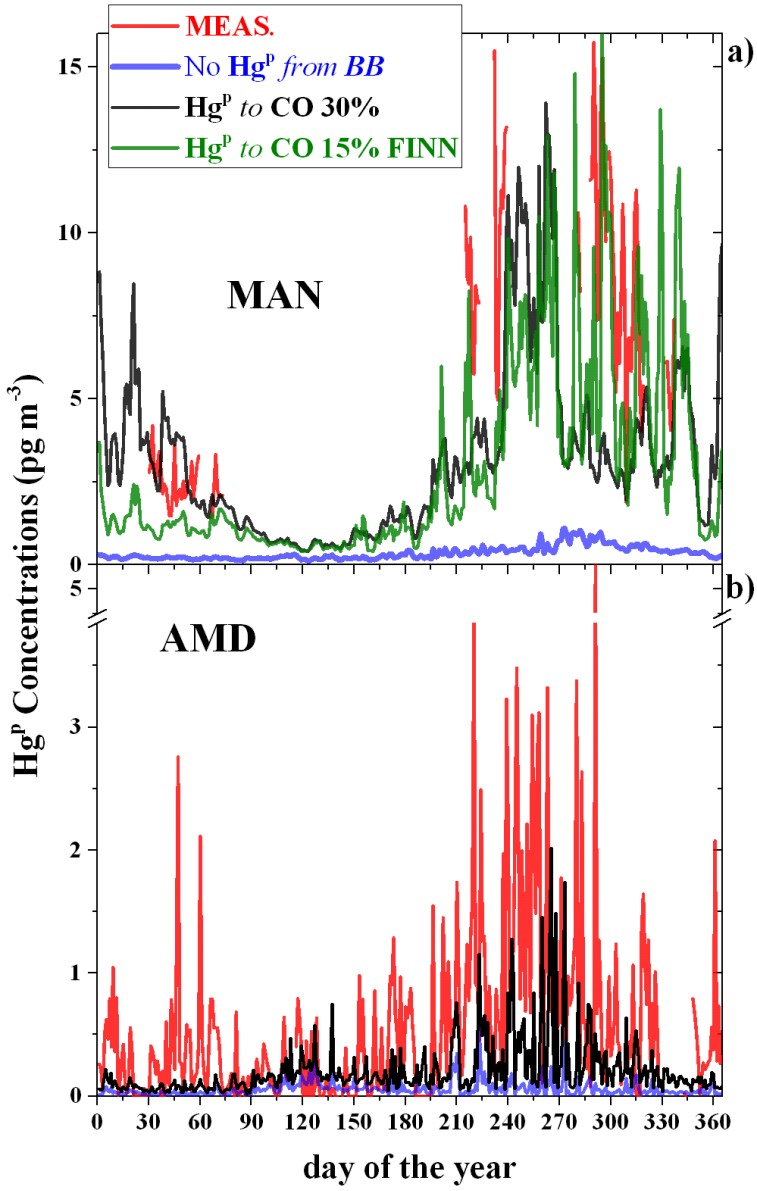

**Figure 10.** Temporal evolution of daily averaged surface Hg$^P$ concentrations measured at Manaus (MAN) and Amsterdam Island (AMD) for the entire 2013, compared with a selection of sensitivity runs.

**Table 1.** Simulations performed

| Name | Inventory (BB emiss Mg) | Full Version | Emiss. Time Res. | Fraction $Hg^P$ | Map $Hg^P$ | Chem.Mech. | Vertical Profile | Scope |
|---|---|---|---|---|---|---|---|---|
| BASE | GFED4.1s (390) | Yes | daily | 15 | CO | O$_3$/OH | PBL | *Reference* |
| 3-hourly | GFED4.1s (390) | | 3-h | 15 | CO | O$_3$/OH | PBL | *Emiss. Time resol.* |
| Monthly | GFED4.1s (390) | | monthly | 15 | CO | O$_3$/OH | PBL | *Emiss. Time resol.* |
| HAM-Profile | GFED4.1s (390) | | daily | 15 | CO | O$_3$/OH | HAM | *Vertical Profile* |
| Only 1st lv | GFED4.1s (390) | | daily | 15 | CO | O$_3$/OH | 1st | *Vertical Profile* |
| Only PBL Lev | GFED4.1s (390) | | daily | 15 | CO | O$_3$/OH | lev of PBL | *Vertical Profile* |
| 3h+HAM-prof | GFED4.1s (390) | | daily | 15 | CO | O$_3$/OH | HAM | *V. Pr. & E. T. res.* |
| $Hg^P$ to PM | GFED4.1s (390) | Yes | daily | 15 | PM | O$_3$/OH | PBL | *$Hg^P$ Mapping* |
| $Hg^P$ to OC | GFED4.1s (390) | Yes | daily | 15 | OC | O$_3$/OH | PBL | *$Hg^P$ Mapping* |
| $Hg^P$ to FMC | GFED4.1s (390) | Yes | daily | variable | CO | O$_3$/OH | PBL | *$Hg^P$ Mapping* |
| NO $Hg^P$ | GFED4.1s (390) | Yes | daily | 0 | NA | O$_3$/OH | PBL | *Fraction $Hg^P$* |
| 4% $Hg^P$ | GFED4.1s (390) | | daily | 4 | CO | O$_3$/OH | PBL | *Fraction $Hg^P$* |
| 30% $Hg^P$ | GFED4.1s (390) | Yes | daily | 30 | CO | O$_3$/OH | PBL | *Fraction $Hg^P$* |
| 100% $Hg^P$ | GFED4.1s (390) | | daily | 100 | CO | None | PBL | *Transport $Hg^P$* |
| Partitioning | GFED4.1s (390) | Yes | daily | 15 | CO | O$_3$/OH | PBL | *Partitioning $Hg^{P/II}$* |
| Partitioning Ref. | GFED4.1s (390) | Yes | daily | 0 | CO | O$_3$/OH | PBL | *Partitioning $Hg^{P/II}$* |
| Reduction | GFED4.1s (390) | Yes | daily | 15 | CO | O$_3$/OH +Red. | PBL | *Chemistry* |
| Br | GFED4.1s (390) | Yes | daily | 15 | CO | Br | PBL | *Chemistry* |
| Br No $Hg^P$ | GFED4.1s (390) | | daily | 0 | NA | Br | PBL | *Chemistry* |
| Br 30% $Hg^P$ | GFED4.1s (390) | | daily | 30 | CO | Br | PBL | *Chemistry* |
| Br $Hg^P$ to OC | GFED4.1s (390) | | daily | 15 | OC | Br | PBL | *Chemistry* |
| Br $Hg^P$ to FMC | GFED4.1s (390) | | daily | variable | CO | Br | PBL | *Chemistry* |
| GFAS | GFASv1.2 (150) | | daily | 15 | CO | O$_3$/OH | PBL | *Inventory* |
| GFAS Br | GFASv1.2 (150) | | daily | 15 | CO | Br | PBL | *Chemistry* |
| FINN | FINNv1.5 (550) | yes | daily | 15 | CO | O$_3$/OH | PBL | *Inventory* |
| FINN Br | FINNv1.5 (550) | | daily | 15 | CO | Br | PBL | *Chemistry* |
| AMAPOH | AMAP2010 | | NA | NA | NA | O$_3$/OH | NA | *Ratio to Anth. Emiss.* |
| AMAPBr | AMAP2010 | | NA | NA | NA | Br | NA | *Ratio to Anth. Emiss.* |
| EDGAROH | EDGAR2008 | | NA | NA | NA | O$_3$/OH | NA | *Ratio to Anth. Emiss.* |
| EDGARBr | EDGAR2008 | | NA | NA | NA | Br | NA | *Ratio to Anth. Emiss.* |
| STREETSOH | STREETS2005 | | NA | NA | NA | O$_3$/OH | NA | *Ratio to Anth. Emiss.* |
| STREETSBr | STREETS2005 | | NA | NA | NA | Br | NA | *Ratio to Anth. Emiss.* |

**Table 2.** Characteristics of ground-based sites measuring Hg$^P$

| Long name | Short name | Lat | Lon | Elev. (m) |
|---|---|---|---|---|
| Amsterdam Island | AMD | -37.8 | 77.58 | 70 |
| Cape Hedo | CHE | 26.86 | 128.25 | 60 |
| Longobucco | LON | 39.39 | 16.61 | 1379 |
| Manaus | MAN | -2.89 | -59.97 | 110 |
| Mauna Loa | MAU | 19.54 | -155.58 | 3399 |
| Mt. Changbai | MCH | 42.4 | 128.11 | 741 |
| Mt. Waliguan | MWA | 36.29 | 100.9 | 3816 |
| Rao | RAO | 57.39 | 11.91 | 5 |

**Table 3.** Horizontal pattern correlation ($R$) and Probabilities that the Hg deposition fields of the different runs belong to the same distribution as the "Base" run ($P_{KS}$). The tick in the Ensemble column indicates the inclusion of the respective run in the Ensemble in Fig. 13

| | Sim. | R | $P_{KS}$ | Ensemble |
|---|---|---|---|---|
| Time resolution | 3-hourly | 1 | 1 | |
| & | Monthly | 1 | 0.99 | |
| Vertical profile | HAM-Profile | 1 | 1 | |
| | 3h+HAM-Profile | 1 | 1 | |
| Hg$^P$ mapping | Hg$^P$ to PM | 1 | 1 | |
| | Hg$^P$ to OC | 1 | 0.42 | ✓ |
| | Hg$^P$ to FMC | 0.99 | 0.45 | ✓ |
| Hg$^P$ fraction | NO Hg$^P$ | 0.94 | 0.38 | ✓ |
| | 4% Hg$^P$ | 0.97 | 0.72 | ✓ |
| | 30% Hg$^P$ | 0.97 | 0.5 | ✓ |
| Inventory | GFAS | 0.98 | 0 | ✓ |
| | FINN | 0.96 | 0 | ✓ |
| Oxidation Mech | Br | 0.96 | 0 | ✓ |
| & | Br No Hg$^P$ | 0.81 | 0 | ✓ |
| Combination | Br 30% Hg$^P$ | 0.91 | 0 | ✓ |
| | Br Hg$^P$ to OC | 0.95 | 0 | ✓ |
| | Br Hg$^P$ to FMC | 0.94 | 0 | ✓ |
| | GFAS Br | 0.94 | 0 | ✓ |
| | FINN Br | 0.92 | 0 | ✓ |

**Table 4.** Hg deposition (Mg) coming from BB to the oceans as obtained by the different runs for the 2013. Last two columns reports the percentage of the total Hg that deposits over seas and lands, respectively.

| | Total Deposition / Mg | | | | | | | | % | |
| Run | N. Atlantic | S. Atlantic | N. Pacific | S. Pacific | Indian Ocean | Med. Sea | Arctic | S. Ocean | SEA | LAND |
|---|---|---|---|---|---|---|---|---|---|---|
| BASE | 31.7 | 32.5 | 75.3 | 67.4 | 45.9 | 1.1 | 5.0 | 2.3 | 66 | 34 |
| NO Hg$^P$ | 32.1 | 32.4 | 82.0 | 74.4 | 48.9 | 1.2 | 4.7 | 2.6 | 71 | 29 |
| 30% Hg$^P$ | 31.3 | 32.5 | 69.3 | 61.0 | 43.2 | 1.0 | 5.2 | 2.0 | 62 | 38 |
| Hg$^P$ to FMC | 31.4 | 32.1 | 74.3 | 66.6 | 44.7 | 1.1 | 5.8 | 2.3 | 66 | 34 |
| Br No Hg$^P$ | 26.6 | 39.4 | 75.8 | 83.0 | 55.3 | 1.1 | 3.7 | 7.6 | 74 | 26 |
| Br 30% Hg$^P$ | 28.0 | 36.4 | 61.7 | 61.1 | 44.9 | 0.9 | 4.8 | 4.6 | 62 | 38 |
| Br Hg$^P$ to FMC | 27.3 | 36.8 | 66.6 | 68.8 | 47.1 | 1.0 | 5.6 | 5.8 | 66 | 34 |

**Table 5.** Mercury deposition (Mg) to the oceans for 2013 from BB and comparison (ratio) with deposition from anthropogenic activities for both oxidation mechanisms.

| O$_3$/OH | N. Atlantic | S. Atlantic | N. Pacific | S. Pacific | Indian Ocean | Med. Sea | Arctic | S. Ocean |
|---|---|---|---|---|---|---|---|---|
| Only BB | 29.8 | 29.9 | 72.1 | 63.0 | 43.0 | 1.1 | 4.7 | 2.1 |
| Only Anthropogenic | 144.0 | 80.0 | 417.7 | 206.7 | 151.3 | 10.0 | 34.3 | 11.0 |
| Ratio | 0.21 | 0.37 | 0.17 | 0.31 | 0.28 | 0.11 | 0.14 | 0.19 |
| Br | N. Atlantic | S. Atlantic | N. Pacific | S. Pacific | Indian Ocean | Med. Sea | Arctic | S. Ocean |
| Only BB | 25.7 | 34.7 | 65.1 | 66.2 | 46.2 | 0.9 | 4.2 | 5.1 |
| Only Anthropogenic | 153 | 85.33 | 457.3 | 188.3 | 140 | 12.33 | 34 | 27.3 |
| Ratio | 0.17 | 0.41 | 0.14 | 0.35 | 0.33 | 0.08 | 0.12 | 0.19 |

**Table 6.** Comparison of the results of BASE and Br simulations including all emissions sources with observations from measurement networks for 2013.

| | Total Gaseous Mercury | | | | Wet Deposition | | | |
| | Regression | | | Stats | Regression | | | Stats |
| | Intercept | Slope | r | NMRSE % | Intercept | Slope | r | NMRSE % |
|---|---|---|---|---|---|---|---|---|
| BASE | 0.36 | 0.62 | 0.72 | 10.54 | 5.84 | 0.04 | 0.12 | 6.89 |
| Partitioning | 0.34 | 0.7 | 0.73 | 11.9 | 3.71 | 0.03 | 0.14 | 4.76 |
| Br | -0.08 | 0.96 | 0.74 | 15.68 | 7.1 | 0.08 | 0.18 | 9.12 |

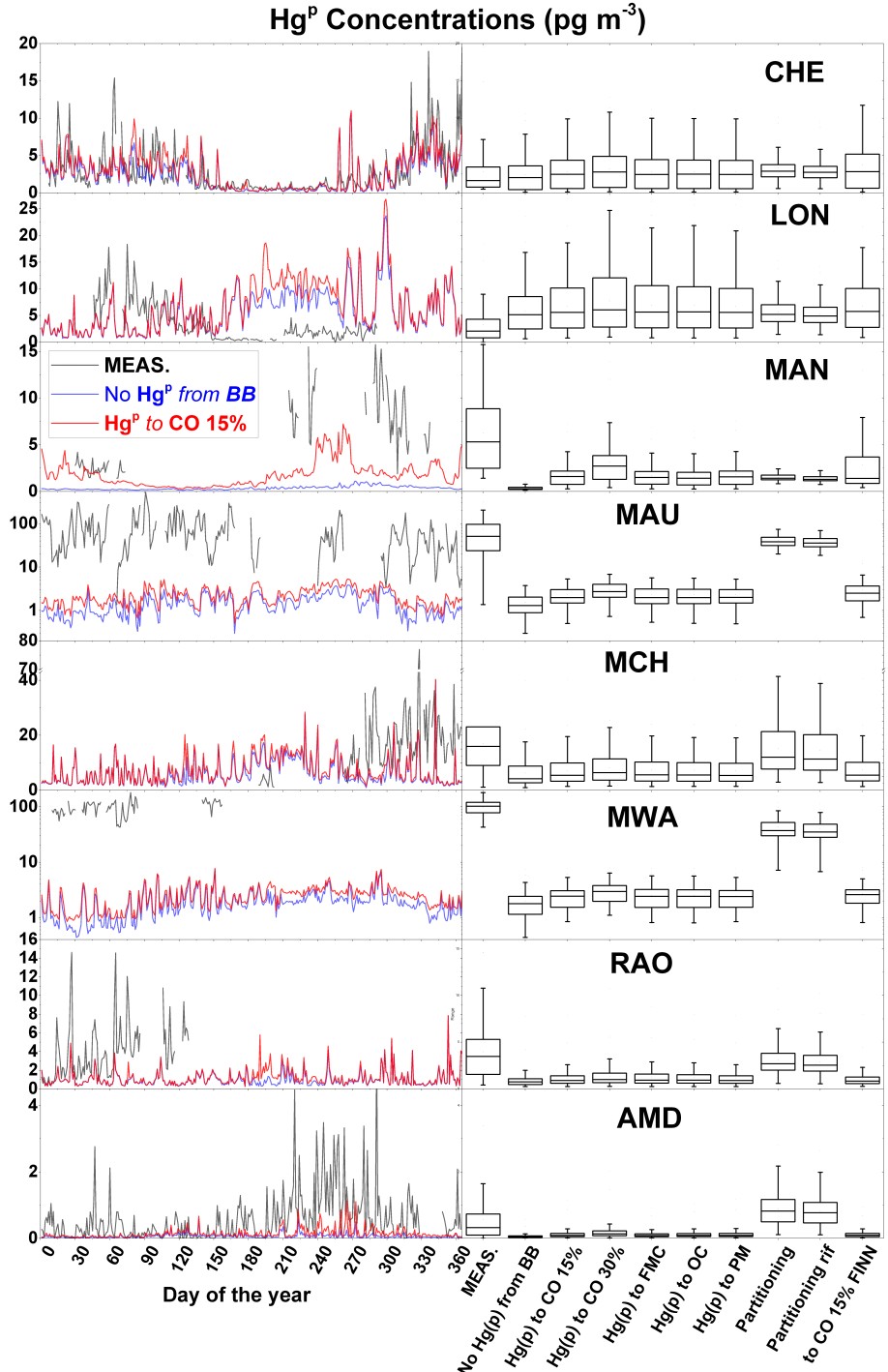

**Figure 11.** (Left column) Temporal evolution of the daily averaged surface Hg[P] concentrations measured at all sites from Table 2 for the entire 2013, compared with the modeled values as simulated by BASE and by NO Hg[P] runs, including emissions from all sources. (Right column) Box plots of the distribution of the of the daily averaged surface Hg[P] concentrations, for the entire 2013, as measured and simulated by the different sensitivity runs. Note the logarithmic for both MAU and MWA subplot

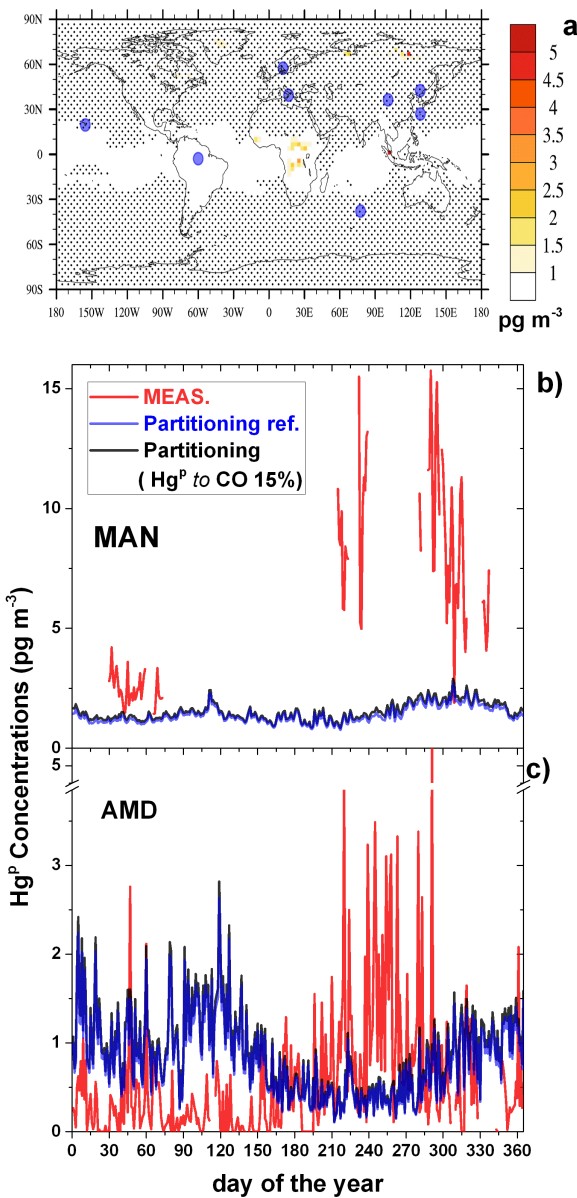

**Figure 12.** (a) Differences in annual averaged surface Hg[P] concentrations as simulated by *Partitioning* and by *Partitioning ref*. runs, both including emissions from all sources and the temperature dependent Hg[II] gas-particle partitioning as implemented in Amos et al. (2012). Black dots indicate that differences are not significant based on a student-t-test at a 95 % confidence interval. Blue bigger points indicate the locations of measurements sites reported in Table 2 Temporal evolution of daily averaged surface Hg[P] concentrations measured at Manaus (MAN) and Amsterdam Island (AMD) for the entire 2013, compared with the modeled values from the same sensitivity runs.

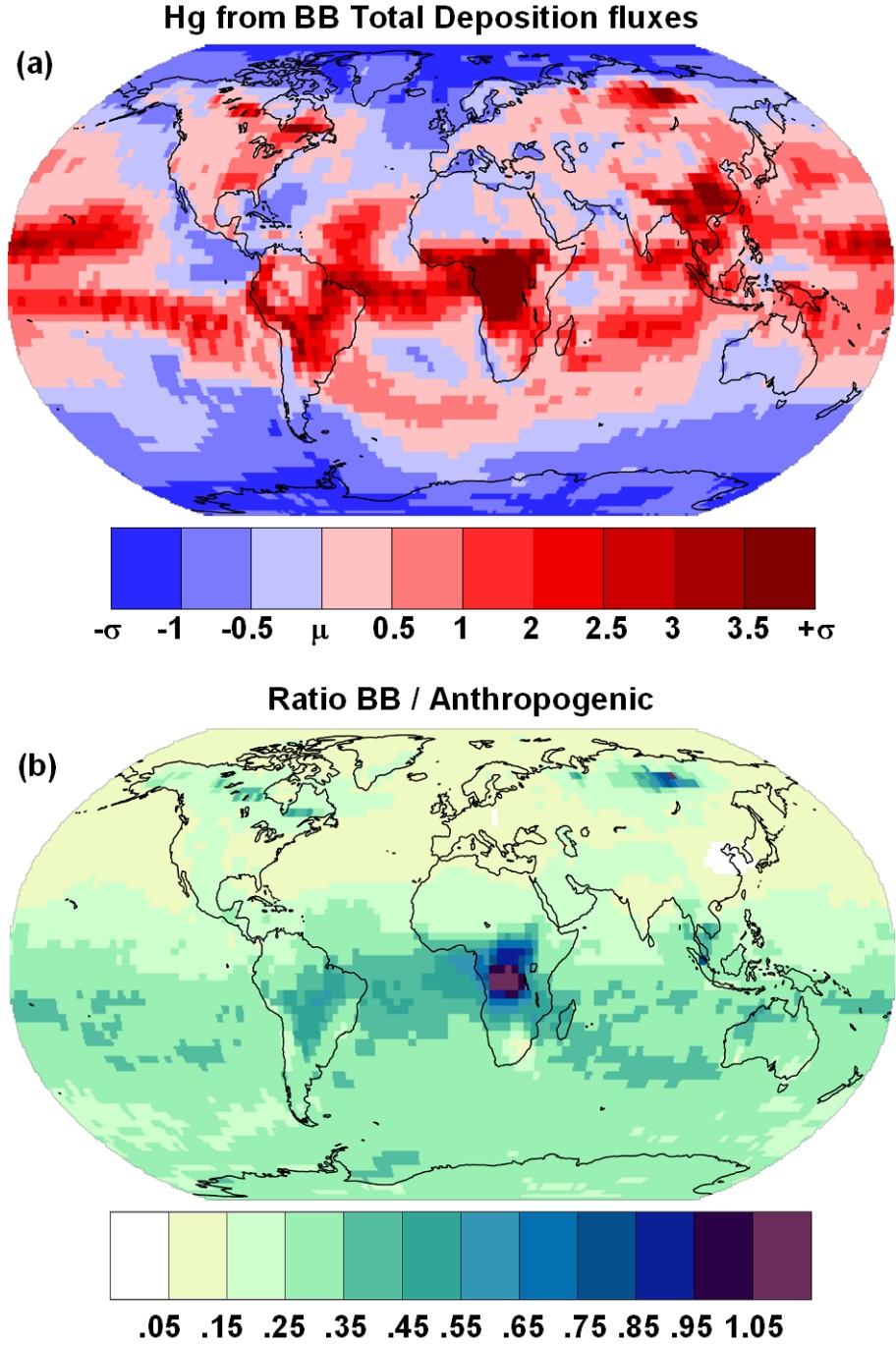

**Figure 13.** Geographical distribution of the total Hg deposition from BB emissions obtained from an ensemble of simulations for the year 2013 (a) in terms of the average ($\mu$) and standard deviation $\sigma$ of the ensemble. The comparison of the BB simulation with an ensemble of runs including only anthropogenic emissions (De Simone et al., 2016) shows (b) the geographic distribution of the fraction of the BB contribution to the Hg deposition from the anthropogenic sources.

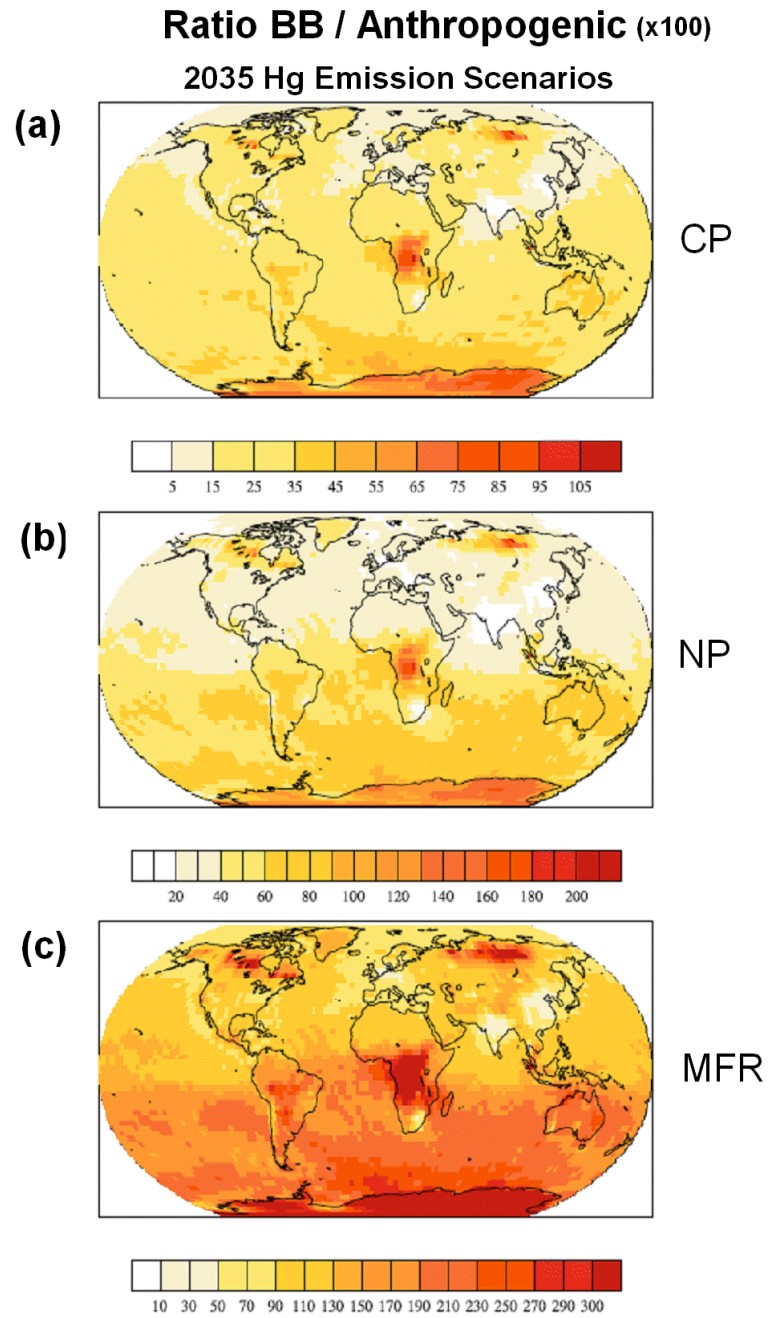

**Figure 14.** Ratio of the Hg deposition due to biomass burning with respect to Hg deposition due to anthropogenic emissions for three anthropogenic emissions scenarios for 2035. (a) CP, Current Policy; (b) NP, New Policy; (c) MFR, Maximum Feasible Reduction.