# Peer review of "Particulate-Phase Mercury Emissions from Biomass Burning and Impact on Resulting Deposition: a Modelling Assessment"

_Atmospheric Chemistry and Physics, 2016_

## Referee Comment (RC1) · Anonymous Referee #1 · 23 Aug 2016

Review of "Particulate-Phase Mercury Emissions during Biomass Burning and Impact on Resulting Deposition: a Modelling Assessment" by Francisco de Simone.

De Simone and co-authors have explored the sensitivity of an atmospheric mercury model (ECHMERIT) to assumptions about mercury emissions from biomass burning. The main focus of their sensitivity tests is the fraction of mercury that is emitted as Hg(p) vs. Hg(0), although they also test model sensitivity to emission time resolution and oxidants for Hg(0). They use several different plausible Hg(p) fractions (0 to 30%) and various way to apportion that fraction (constant or proportional to biomass burning CO, PM, or OC). The partitioning of emissions is an important issue, as the authors explain, because Hg(0) has a long atmospheric residence time and circulates globally

while Hg(p) has a short residence time and deposits near the emission source. These are reasonable sensitivity tests and I expect that other mercury scientists and modelers will be interested in the results.

The main weakness of the paper is that it provides no comparison to observations, except for an unexplained and unused table in the Appendix. The paper therefore provides no insight into which, if any, of the many model configurations provide reasonable comparisons with observations. There are abundant surface and aircraft measurements of Hg(0), Hg(p), CO, OC, and PM that could be used for this purpose (GMOS, AMNET, CARIBIC, ARCTAS, INTEX-B). If the ECHMERIT model is run in a climate mode, so that it does not match the daily weather conditions at measurement sites, the simulated distributions and correlations between multiple species can still be compared with observations. Without comparison between model and observations, I do not think that this paper in its current form is suitable for publication in ACP.

Another significant problem with the current version of the manuscript is that the methods do not contain enough detail to understand how the emissions were constructed. How are biomass burning emissions of Hg (=Hg(p)+Hg(0)) calculated from the biomass burning CO or DM provided by GFED? Please provide the relevant emission factors or emission ratios. In simulations where Hg(p) fraction depends on OC, PM, or FMC the manuscript needs to clearly explain how the Hg(p) fraction is calculated from OC, PM, or FMC. Do the emission factors (e.g. CO/DM, OC/DM) vary geographically with biome type? A simulation with 100% Hg(p) from biomass burning is discussed in Sect. 3.3 but not described in the methods. Regardless of how Hg is calculated in the emission inventory, please report the Hg/CO ratio because this would enable comparison to many observations that are reported this way.

Other issues Is there chemical reduction in the model? If so, is Hg(p) affected by it?

The total Hg emissions from biomass burning in this work are 400 Mg/yr. A previous analysis by the same authors reported much higher mean emissions of 675 Mg/yr

(Pirrone et al., 2010). What is the reason for such a large change?

None of the figures show the spatial map of Hg(p)/Hg ratio (or Hg(p)/Hg(0) ratio), which is the central focus of the paper. In addition, all 4 panels of Figure 2 are visually indistinguishable (and indistinguishable from Fig 1, except for magnitude). I think this space would be better used to show the Hg(p)/Hg emission ratios under the various schemes based on CO, PM, and OC.

Some additional observational studies of Hg in biomass burning plumes should be discussed: Ebinghaus et al., 2007; Holmes et al., 2010.

Page 1 Line 3 (P1L3): Add that the Hg which is not Hg(p) is assumed to be Hg(0).

P1L13: 71% to 62% of what?

P1L15: Statement about mercury in water-stressed and warming forests is speculation that is not supported in the paper.

P1L19: Statement exaggerates the magnitude of biomass burning emissions relative to other anthropogenic emissions; it is certainly less than 1/2 of anthropogenic Hg emissions. First, it is widely acknowledged that a very large portion of biomass burning is anthropogenic, even though emission inventories are not labelled this way. Second, the Muntean et al., 2014 paper does not include mercury emissions from small-scale gold mining, so anthropogenic emissions are much larger than they estimated.

P3L2: particle emissions are presumably also calculated.

P4L2: "of" great importance

P4L9: Define FMC

P4L23: Is the total Hg emission the same in all simulations? How is it calculated from the GFED DM or CO?

P4L23. "Considering" should begin a new sentence.

P5L14: How are the data in the figures normalized?

P5L16: It seems very unlikely that the fairly smooth zonal-mean distribution would be altered by finer spatial resolution.

P6L23: What does "passive tracer" mean? In atmospheric modeling, "passive" usually means that a tracer does not alter the model's transport or physics. (i.e. it is passively transported.) I would therefore expect that all of the Hg species in all of the simulations are passive in this sense.

P6L29: Not quite correct. Oxidant choice still has a big effect on the deposition pattern.

P7L1: Statement says that vertical profile of emissions doesn't matter, but I expect that the vertical profile would be quite important for scenarios with high Hg(p) emission fraction. Like other aerosols and reactive gases, Hg(p) emitted into the free troposphere should disperse much farther than Hg(p) emitted into the boundary layer.

P7L6: 66% of what?

Fig 7c: Panel title says "Hg(p) fraction =30%" but one of the plotted quantities is "100% Hg(p)". Only one can be correct.

Table 2: How are the correlations calculated? Are they the spatial correlation of the annual mean? Is temporal variability considered in the correlations? What does "Ensemble" mean here?

Table 3: Title should say "from biomass burning"

Table 4: Title says "Mercury deposition (Mg)" but only some rows have units of Mg.

Table 5: Terms "BASE Full" and "Br Full" are not defined.

References Holmes, C. D., Jacob, D. J., Corbitt, E. S., Mao, J., Yang, X., Talbot, R. and Slemr, F.: Global atmospheric model for mercury including oxidation by bromine atoms, Atmos. Chem. Phys., 10(24), 12037–12057, doi:10.5194/acp-10-12037-2010,

ACPD

2010.

Ebinghaus, R., Slemr, F., Brenninkmeijer, C. A. M., van Velthoven, P., Zahn, A., Hermann, M., O'Sullivan, D. A. and Oram, D. E.: Emissions of gaseous mercury from biomass burning in South America in 2005 observed during CARIBIC flights, Geophys. Res. Lett., 34(8), doi:10.1029/2006GL028866, 2007.

Pirrone, N., Cinnirella, S., Feng, X., Finkelman, R. B., Friedli, H. R., Leaner, J., Mason, R., Mukherjee, A. B., Stracher, G. B., Streets, D. G. and Telmer, K.: Global mercury emissions to the atmosphere from anthropogenic and natural sources, Atmos. Chem. Phys., 10(13), 5951–5964, doi:10.5194/acp-10-5951-2010, 2010.

---

## Referee Comment (RC2) · M. Cohen (Referee) · 3 Sep 2016

Review of "Particulate-Phase Mercury Emissions during Biomass Burning and Impact on Resulting Deposition: a Modelling Assessment", by De Simone et al., Atmos. Chem. Phys. Discuss.

GENERAL COMMENTS

De Simone et al. present a model-based study of mercury emissions and deposition arising from global biomass burning (BB), examining a range of different model inputs and assumptions, with particular emphasis on the fraction of mercury emitted from BB as particulate mercury [HgP]. Overall, this seems to be an excellent investiga-
tion, although as noted below, there are some areas that might need some additional explanation and/or justification.

SPECIFIC COMMENTS

1. The model year 2013 was selected. How does 2013 compare with other years in terms of BB emissions? Should note that conclusions from this work apply to 2013 and will likely be at least somewhat different for different years.

2. (P2.L27). Would be helpful if you could say something about the differences in the inventories. E.g., a few sentences at least regarding the essential differences in how they were constructed, and of course, what the different emissions were in each inventory.

3. Section 2.2 Experimental Setup. Would be helpful if you included here (or elsewhere) additional details about the model. Here are some details, for example, that might be helpful:

* Is HgP created from Hg(0) oxidation in the model, and if so, what fraction of the oxidation products are assumed to be HgP with different reactions, etc.?

* Once HgP is emitted into the model (and/or created within the model), can it be transformed to any other form of mercury, i.e., can HgP be converted to Hg0 or Hg2 in the model?

* Is there any conversion or partitioning of Hg2 to HgP in the model? If so, how is this estimated, and is it reversible?

* What particle size(s) are assumed for HgP? What information exists on the particle size distribution of HgP in the BB plumes? This would seem to be a very important factor, considering particulate deposition is critically influenced by particle size. This could be noted as a relatively uncertain aspect of the simulation that is not being addressed in the present study.

[Figure]

* How is particulate dry deposition handled in the model? Is gravitational settling velocity factored in? If so, what are the size(s), shape factor(s), and density(s) of HgP-carrying particles?

* How is particulate wet deposition handled in the model? In my modeling work, I have found the parameterisations used in HgP wet deposition to have a very big impact on the fate/transport of HgP.

* Has the model been evaluated by comparison against HgP measurements? If so, what were the results?

4. The model is being run with a relatively coarse grid (e.g., on the order of 2.8 x 2.8 degrees at the equator), and so, as with any model of this type, sub-grid phenomena could be adding uncertainty to the results. Especially, for example, for emissions from BB, the height of emissions could significantly impact the near-field deposition. In real-world BB situations, the emissions will not be uniformly distributed throughout the PBL, and deposition from the real vertical distribution could be much different than that with the assumed uniform-PBL assumption. In some cases, the near-field deposition could be much greater, to the extent that the emissions are emitted nearer to the ground. Along these same lines, the authors do carry out a simulation with emissions confined to the first layer of the PBL. While the height of this layer does not appear to be specified in the paper, I'm not sure it should be considered such an unrealistic simulation, as is done in the analysis. The fact that it seems to give relatively different results could be seen as evidence that emission height really does make a difference. While I am not that familiar with the literature, I believe there have been numerous studies published regarding the height of BB emissions under different conditions. As a related point, the manuscript notes that "In particular high HgP fractions were observed during smouldering phases, whereas very low or undetectable HgP levels were found during flaming combustion." [P4.L18-19]. This could mean that the highest HgP emissions might occur with relatively low injection heights, i.e., if the injection heights under smouldering conditions are lower than the heights under more intense combustion conditions.

5. Figures 6 and 7 are a really interesting way to present the results! However, it took a little time to get my head around what they were saying at first. Perhaps a little more explanation could be added in the caption for these figures?

Technical corrections and/or suggestions

(...Note that in the following, if a wording change or other correction is being suggested, I have simply included the final wording being suggested, rather than any sort of "track changes" notation. Apologies if this leads to any lack of clarity.)

P1.L22. "Its relative importance may increase in the coming years, e.g., if the Minimata Convention and/or other efforts lead to reductions in anthropogenic emissions."

P2.L16-17. "...resulting from BB, when variations in HgP fractions and production processes are considered."

P2.L17-19. "The most recent version of the GFED BB emission inventory (van der Werf et al., 2010; Randerson et al., 2012; Mu et al., 2011), has been included in the global online Hg chemical transport model ECHMERIT, to simulate Hg deposition from BB for the year 2013 and to quantify the influences of variations in model inputs, assumptions and parameterisations."

P2.L23. "... version of the inventory..."

P2.L24. Need period at end of sentence.

P2.L27. Wouldn't these be considered "sensitivity" runs, rather than "control" runs?

P2.L27. "... see Andela et al. (2013) (and references therein) for a description..."

P3.L4. "Unless explicitly stated,..."

P3.L9. "This value is within the range of observations (Obrist et al., 2007; Finley et al., 2009). However, since there are uncertainties in Hg speciation from BB (Zhang et al., 2013), further simulations were carried out with varying fractions of HgP ( 0%, 4% and

30%)."

P3.L15-17. "The principal vertical profile used (PBL-Profile) maps Hg emissions uniformly within the Planetary Boundary Layer (PBL), whereas in the second, the vertical profile of the standard version of the ECHAM-HAM model was used (HAM-Profile)(Zhang et al., 2012)."

P3.L16. Could the "HAM" acronym be defined the first time it's used?

P3.L21-23. "These simulations primarily employ a O3/OH Hg0(g) oxidation mechanism. However, since the precise atmospheric Hg oxidation mechanism remains unclear (Hynes et al., 2009; Subir et al., 2011, 2012; Gustin and Jaffe, 2010; Gustin et al., 2015), a number of runs were performed using a Br-based oxidation mechanism."

P3.L28-29. "Finally two simulations were conducted including Hg emissions from all sources and including re-emissions, to evaluate model performance against measurements (see Appendix A)."

P3.L28-29. What additional emissions were used for these "all-source" simulations?

P3.L32. "The majority of Hg releases from BB is believed to occur as Hg0(g)."

P4.L7. "properties" is misspelled.

P4.L13-16. What equation(s) were used, with what parameters? That is, you say that the Hg0 to HgP ratio is determined by FMC, but what is the mathematical relationship used?

P4.L27. I cannot really see very many "notable differences" in Figures 1 and 2. Part of the issue is that the figures are very small and the color ramp does not have a lot of contrast. Could the figures be bigger?

P5.L3. At a number of points in the document, it is stated that only the 85:15 emissions speciation results are shown "for clarity". Its not clear to me why showing the results for other speciation profiles would make things less clear. There would be more figures,

but would clarity really suffer?

P5.L14. How were the latitudinal deposition profiles normalized?

P5.L18. What is the height of the first model level?

P5.L20-22. A few comments about the following sentence: "This last vertical distribution scenarios are unrealistic, however the differences obtained here contrast with the findings of De Simone et al. (2015) and are due to the fraction of HgP included in this study."

(a) Not exactly sure what you are trying to say here in terms of comparison to findings of De Simone et al. (2015).

(b) As noted above in the Specific Comments, I'm not sure I agree that the vertical distribution being referenced is unrealistic.

(c) This sentence needs to be reworded somewhat for grammar and clarity.

P5.L28. Do you mean the "deposition peak"?

P6.L1. Maybe would be clearer if the section was called something like this: "Impact of atmospheric oxidation pathway and speciation profiles on geographic distribution of deposition".

P6.L10-12. This sentence is a little confusing, particularly with the use of "all" towards the end. This "all" confused me before I realized you didn't really mean "all".

"To better understand the combined effect of Hg speciation and oxidation pathway on the deposition distribution, agreement maps were created, to highlight the model cells where different simulations all predict significant deposition..."

Maybe better to say something like this:

"To better understand the combined effect of Hg speciation and oxidation pathway on the deposition distribution, agreement maps were created, to highlight the similarities

and differences in the distribution of high-deposition model cells in different simulations..."

P6.L12. What statistical distribution is the "standard deviation" calculated for, e.g., is it the combined data set of cell-by-cell deposition for all cells in all relevant simulations?

P6.L14. "Using the O3/OH mechanism, the number of model cells in which the model predicts high deposition..."

P6.L21. maybe "contrasts" (or simply "presents") rather than "confronts"

P6.L22. Not sure what you mean by "passive tracer" in this context. It still deposits, right? In other simulations, how are HgP emissions not like a "passive tracer" in the same context? I guess you are implying here that there is no chemical reactions in which Hg0 is oxidized to HgP, and/or that there are no processes converting HgP to another form of Hg. And so, there should be no impact of the oxidation mechanism chosen. But, as noted above in the specific comments above, you could add some additional detail to the text regarding these and other issues to make things clearer.

P6.L26. Seems like maybe this section could be divided into two. One called "Uncertainty" and one called "Biomass Burning versus Anthropogenic Impact"

P7.L1. Could refer the reader to the figure or table that shows the point you are making. Also, instead of "actually have no influence", could say something like "have little influence". And as noted above, you haven't convinced me that the emissions into the first model level – or at least emissions into something less than the full PBL – are really "unrealistic".

P7.L7. I don't see the Antarctic in the tabular results, but you give results here?

P7.L12. "... as in De Simone et al. (2015)." (and same correction a few lines later)

P7.L13-16. What is an "inspected ensemble"? How was the eventual ensemble created – medians of values for each cell, or mean values for each cell, or some other

method?

P7.L24. "just about everywhere" (seems like there are a few locations less than 25%?)

Table 1. Model Evaluation (not Model Validation)

Table 2. Would be helpful to explain the "R" and "P-KS" parameters a little either in the Table or in the text. At least to me, it seems a little too cryptic.

Table 3 and Table 4. Maybe could make these into some sort of graphic, either instead of or in addition to?

Table 5. What measurement sites? How many sites? What networks? What averaging time for "r" and for "NMRSE%"? Need some more detail here. What about comparison against HgP measurements? This would seem to be important for this paper!

Figure 3, and in fact, most figures: Why so small? For Figure 3, could make it much wider and I think would be much clearer. Difficult to see data when lines overlap so much. Maybe consider some sort of differential dotted/dashed line(s) so that they might be able to be distinguished even when congruent?

---

## Referee Comment (RC3) · Anonymous Referee #3 · 20 Sep 2016

De Simone et al. (doi:10.5194/acp-2016-685) provide a very detailed model sensitivity study on the influence of partitioning of particulate mercury from biomass burning on its deposition patterns. Such partitioning effect has not been incorporated into most mercury chemical transport models, but it is worthy of attention in the mercury community. The topic of this study is well within the scope of ACP. However, I think the authors should address the following general and specific comments before its consideration of publication.

General comments: (1) A major weakness of this manuscript is lack of modelobservation comparison. The authors point out several significant differences of the deposition fluxes in different model scenarios. Do the available observations provide

constraints on the parameterizations of mercury BB emissions? (2) It has been suggested that the partitioning of mercury in the atmosphere depends on temperature and aerosol concentrations (for example, Amos et al., 2012). What is the treatment in this study and what is its scientific basis? (3) More details of the model parameterizations should be provided. A key process is the photo-reduction of oxidized mercury in the atmosphere. Does the model allow such process in this study? Would this process affect the major conclusion of this study?

Specific comments: (1) Title: I suggest changing "during" to "from". (2) Page 2, line 31: What is the global average enhancement ratio? Does it fit in the observed range (for example, Slemr et al., 2014)? (3) Page 3, lines 15-20: I do not quite understand why these two schemes of vertical profiles are equal less than 4 km. Could more explanations be given here? (4) Sect. 2.4: Are there any statistical relationships among OC, PM, and FMC? I am curious since they are all linked to the combustion characteristics. (5) Page 4, line 29: Could more explanations be given about the differences of the emission (and also the deposition) patterns > 60-degree north in difference scenarios (mapping to OC vs FMC)? (6) Figure 4: It seems that the influences of different parameterization of PBL-type vertical profiles and different temporal resolutions are insignificant. Could these be due to the gross spatial and temporal resolutions of the model used in this study?

Editorial comments: (1) Page 1, line 17: add brackets for "Hg". (2) Page 1, line 23: "asses" should be "assess". (3) Page 2, line 6: add a comma before "however". (4) Page 2, line 27: wrong reference format. (5) Page 4, line 2: "is of great importance". (6) Page 4, line 28: "emissions". (7) Page 5, line 11: "where" should be "were". (8) Page 6, line 28: remove comma. (9) Page 7, line 12, 15: wrong reference format. (10) Page 8, line 14: remove "the"; full name of "TGM".

Reference: Amos, H. M., Jacob, D. J., Holmes, C. D., Fisher, J. A., Wang, Q., Yantosca, R. M., Corbitt, E. S., Galarneau, E., Rutter, A. P., Gustin, M. S., Steffen, A., Schauer, J. J., Graydon, J. A., Louis, V. L. St., Talbot, R. W., Edgerton, E. S., Zhang, Y., and
Sunderland, E. M.: Gas-particle partitioning of atmospheric Hg(II) and its effect on global mercury deposition, Atmos. Chem. Phys., 12, 591-603, doi:10.5194/acp-12-591-2012, 2012. Slemr, F., Weigelt, A., Ebinghaus, R., Brenninkmeijer, C., Baker, A., Schuck, T., Rauthe-Schöch, A., Riede, H., Leedham, E., Hermann, M. and van Velthoven, P., 2014. Mercury plumes in the global upper troposphere observed during flights with the CARIBIC observatory from May 2005 until June 2013. Atmosphere, 5(2), pp.342-369.

---

## Referee Comment (RC4) · T. Dvonch (Referee) · 22 Sep 2016

Review of "Particulate-Phase Mercury Emissions during Biomass Burning and Impact on Resulting Deposition: a Modelling Assesment" by Francisco De Simone et al.

General Comments:

De Simone et al. present a detailed assessment of the impact of mercury emissions from biomass burning on resulting atmospheric mercury deposition. The assessment is completed through utilization of an updated emissions database and an updated global mercury chemical transport model. Within this framework, the authors investigate a variety of model parameterizations and the role of other uncertainties on resulting magnitudes and spatial distributions of mercury deposition. Overall, this work represents a sizable effort and further informs the scientific community regarding speciation of mercury emissions, chemical oxidation mechanisms, and spatial and temporal variations, all within the specific context of biomass burning and with relevance to the implementation of mercury policy.

This manuscript represents a substantial contribution to the field, and is very much with the scope of ACP. However, there are several items that could be addressed by the authors and incorporated into revisions that would likely strengthen the manuscript overall.

Specific Comments:

1. Much of the manuscript focuses on the potential impact of Hg P emissions. However, the issue of particle size is not discussed in the paper. Certainly, there have been assumptions made within the model regarding particle size, with direct implications to the potential transport distance prior to Hg P removal from the atmosphere (via either dry or wet processes). The manuscript would benefit from added discussion specific to particle size.

2. Table 5 presents summary statistics regarding comparison of model output with available observations from measurement networks. However, there is little text in the body of the manuscript in support of the inclusion of Table 5. The manuscript would benefit from added discussion to characterize and specify how well the model performed in comparison to observations.

3. No explanation or justification is provided on the selection of 2013 as the model time period. This rationale should be provided in the revisions, along with some indication of the representativeness of 2013 compared to other recent years.

4. Figures 1, 2, 5, & 9 seem to be very instructive. However, they are not easily legible. The size/resolution of these figures should be improved for the benefit of the reader.

Technical/Editorial Comments:

The units reported in Table 3 need additional clarification (Mg and %).

Page 1 - line 13, "71% to 62%"…of total deposition? Seems this sentence is missing some needed context.

Page 2 - line 7, should be "fraction of Hg emitted" (add "of").

Page 2 – line 24, period needed after "(Randerson et al., 2012)".

Page 3 – line 8, "equal to the 15%" (remove "the").

Page 4 – line 2, "Hg emissions is of great importance" (add "of").

Page 5 – line 18, "first model level level leads to" (remove "level").

Page 5 – line 19, "approx" should be "approximately".

Page 7 – line 1, instead of "have no influence", perhaps "have little influence"?

Page 8 – line 14, "between the the measurement" (remove "the").
* * *

---

## Author Comment (AC2) · 22 Dec 2016

M. Cohen (Referee)

mark.cohen@noaa.gov

Received and published: 3 September 2016

Review of "Particulate-Phase Mercury Emissions during Biomass Burning and Impact

on Resulting Deposition: a Modelling Assessment", by De Simone et al., Atmos. Chem.

Phys. Discuss.

**GENERAL COMMENTS**

De Simone et al. present a model-based study of mercury emissions and deposition arising from global biomass burning (BB), examining a range of different model inputs and assumptions, with particular emphasis on the fraction of mercury emitted from BB as particulate mercury [HgP]. Overall, this seems to be an excellent investigation, although as noted below, there are some areas that might need some additional explanation and/or justification.

We thank the referee for his positive general comments and also for his specific comments and feedback that helped us to improve the general quality of the manuscript.

**SPECIFIC COMMENTS**

1. The model year 2013 was selected. How does 2013 compare with other years in terms of BB emissions? Should note that conclusions from this work apply to 2013 and will likely be at least somewhat different for different years.

The authors have already investigated many uncertainties related to Hg emissions from BB in De Simone et al., 2015, including the year-to-year Hg BB emission variability for a decade. As explained above, this study focusses on the speciation of Hg emissions from BB, and on the effects on the resulting deposition, and it is investigated for the first time in a CTM. Results for other years could be somewhat different. However we decided to choose 2013 because it was one of the years best covered by measurements within the GMOS project. This allows us to have feedbacks from the comparison with measurements collected at a global scale.

**We modified the text to include the reason for the choice:**

This study cover a single year, the 2013, which has been chosen due the large availability of measurements from GMOS network \citep{Sprovieri2016\_conc, Sprovieri2016\_wet,Damore2015}.

These results apply for the investigated year (2013) and could be to some extent different considering other years, due to the complex interaction of the numerous actors determining the final fate of \ce{Hg}. However few alternatives of analysis period exist due the limited time coverage of global measurement network(s).

2. (P2.L27). Would be helpful if you could say something about the differences in the inventories. E.g., a few sentences at least regarding the essential differences in how they were constructed, and of course, what the different emissions were in each inventory.

**The inventories used for this study, GFAS, GFED and FINN, and the differences about the way they are compiled, are fully detailed in Andela et al. 2013, and also partially in De Simone et al., 2015. However, in the revised text we added some details and a column in the Table 1 reporting the total amount of Hg emissions from BB included in each run/inventory.**

These three inventories are all compiled using the imagery obtained from the MODIS instruments. However, the way by which the data are filtered or processed yields to substantial differences among the final product, see \citet{Andela2013} and references therein for a detailed description of the differences among the inventories.

3. Section 2.2 Experimental Setup. Would be helpful if you included here (or elsewhere) additional details about the model. Here are some details, for example, that might be helpful:

**We extended the relevant sections to describe better the parameterizations included in the model, either in the base configuration or in the variants considered.**

\* Is HgP created from Hg(0) oxidation in the model, and if so, what fraction of the oxidation products are assumed to be HgP with different reactions, etc.?

\* Once HgP is emitted into the model (and/or created within the model), can it be transformed to any other form of mercury, i.e., can HgP be converted to Hg0 or Hg2 in the model?

\* Is there any conversion or partitioning of Hg2 to HgP in the model? If so, how is this estimated, and is it reversible?

In the base configuration of the model Hg(p) is assumed to be inert, it is not considered a product of Hg(0) oxidation. It is emitted from either anthropogenic or BB (if any) sources, and it is subject to transport and deposition processes only. However some studies (Steffen et al., 2014, Amos et al, 2012) have been suggested that a partitioning of reactive Hg (i.e., Hg(II)) between gas and particle might exist. In particular has been suggested that it could be driven by air temperature and availability of aerosol particles (Amos et al, 2012). Therefore, two other simulations were conducted including this temperature dependent gas-particle partitioning, to assess the impact of considering a fraction of Hg from BB as Hg(p) under this assumption.

The atmospheric reduction of Hg(2) to Hg(0) has been included in different modeling studies, including De Simone et al, 2014, to regulate the atmospheric residence time of elemental Hg and to finally best match the observations. The mechanisms that have been proposed are many, including the photoreduction of the oxidized Hg. However some of them are unlikely to occur under most atmospheric condition. Due to these uncertainties, we preferred to not include reduction in this study.

**We modified the text opportunely:**

 $ce{Hg^{P}}$  is assumed to be inert, whenever it is emitted from anthropogenic or BB activities, is subject to transport and deposition processes and it is not involved in any chemical reactions.

•••

Some studies \citep{Steffen2014,Amos2012} suggested that the partitioning of reactive specie between gas and particle might be driven by air temperature and on availability of aerosol particles. Therefore, two other simulations were conducted including the temperature dependent gas particle partioning described \citet{Amos2012}, one assuming BB \ce{Hg} emissions to be only  $ce{Hg^{0}_{0}}$ , and another assuming a 15\% of BB \ce{Hg} emissions as  $ce{Hg^{P}}$ .

Atmospheric reduction of  $ce{Hg^{II}_{(g/aq)}}$  to  $ce{Hg^{0}_{(g)}}$  has been included in many models to regulate the residence time of  $ce{Hg^{0}_{(g)}}$  in the atmosphere. However, a number of the proposed mechanisms are unlikely to occur under most atmospheric conditions, or are based on empirical rates to better match the observations, see  $citet{Kwon2016}$  for a recent review. Due to this uncertainty, reduction was not included in this study.

\* What particle size(s) are assumed for HgP? What information exists on the particle size distribution of HgP in the BB plumes? This would seem to be a very important factor, considering particulate deposition is critically influenced by particle size. This could be noted as a relatively uncertain aspect of the simulation that is not being addressed in the present study.

The particle size distribution has undoubtedly an impact on the final fate of Hg(p) emitted by different sources. However there are large uncertainties regarding the size distribution of particles emitted, and how it evolves during the different phases of BB (see for example, Janhäll and Pöschl, 2010 and the reference therein).

**In the revision paper we included the following text:**

No further  $ce{Hg^{p}}$  particle dimension distributions other than the standard log-normal particle size distribution, as described in detail in  $citep{Jung2009}$ , were considered in this study due to large uncertainties regarding the dynamic size range of particle emitted during BB, see  $citet{Janhall2010}$  and the references therein.

\* How is particulate dry deposition handled in the model? Is gravitational settling velocity factored in? If so, what are the size(s), shape factor(s), and density(s) of HgP-carrying particles?

\* How is particulate wet deposition handled in the model? In my modeling work, I have found the parameterisations used in HgP wet deposition to have a very big impact on the fate/transport of HgP.

Dry deposition velocities are calculated considering both dry deposition and gravitational settling, following Slinn and Slinn (1980), and similar to the implementation within the CAMx model (CAMx, 2006). The assumed log-normal particle size distribution is divided into a fixed number of size intervals, then the deposition velocity is calculated for each interval and finally these are aggregated in a weighted mean.

Regarding the wet deposition of different species, both below-cloud and in-cloud scavenging are considered. Wet scavenging of dry particles only occurs below precipitating clouds and it is proportional to the mixing ratios of air pollutants. The scavenging rate, depends on scavenging efficiency, total rainfall intensity, a mean cloud or rain droplet radius and rain droplet falling velocity, following the approach of Seinfeld and Pandis (1998), and similar to the implementation within the CAMx model (CAMx, 2006).

All these mechanism remain unchanged in the model since Jung et al., 2009, where they are described in detail. Therefore we prefer not to include too much detail in this study, and to refer to Jung et al., 2009.

**In the revision paper we included the following text:**

Mechanisms and parameterizations used for calculating the dry and the wet deposition of the different \ce{Hg} species are the same as described by \citet{Jung2009}.

\* Has the model been evaluated by comparison against HgP measurements? If so, what were the results?

In the revised text we have included a new subsection within section 3 dedicated to the comparison with Hg measurements from the GMOS network for 2013, to validate the model, and to assess any feedbacks/constraints related to the different assumptions considered about the Hgp emissions from BB. More particularly, when considering the Hg emissions from all other sources, the very small perturbation produced by moving a fraction of Hg BB emissions from Hg0 to Hgp in almost all sensitivity runs causes very little perturbation to the TGM and wet deposition results. Conversely the Hgp in air concentration samples collected in a number of sites from GMOS networks for the year 2013 enabled us to assess the impact of Hgp emissions from BB and to distinguish between the different assumptions. In particularly at two remote sites the model runs including a fraction of Hg(p) from fires resulted in a better agreement with measurements.

We included the new Section 3.4 "Constraints from Global Measurements networks" see page 7 of the revised paper.

4. The model is being run with a relatively coarse grid (e.g., on the order of 2.8 x 2.8 degrees at the equator), and so, as with any model of this type, sub-grid phenomena could be adding uncertainty to the results. Especially, for example, for emissions from BB, the height of emissions could significantly impact the nearfield deposition. In real world BB situations, the emissions will not be uniformly distributed throughout the PBL, and deposition from the real vertical distribution could be much different than that with the assumed uniform-PBL assumption. In some cases, the near-field deposition could be much greater, to the extent that the emissions are emitted nearer to the ground. Along these same lines, the authors do carry out a simulation with emissions confined to the first layer of the PBL. While the height of this layer does not appear to be specified in the paper, I'm not sure it should be considered such an unrealistic simulation, as is done in the analysis. The fact that it seems to give relatively different results could be seen as evidence that emission height really does make a difference. While I am not that familiar with the literature, I believe there have been numerous studies published regarding the height of BB emissions under different conditions. As a related point, the manuscript notes that "In particular high HgP fractions were observed during smouldering phases, whereas very low or undetectable HgP levels were found during flaming combustion." [P4.L18-19]. This could mean that the highest HgP emissions might occur with relatively low injection heights, i.e., if the injection heights under smouldering conditions are lower than the heights under more intense combustion conditions.

We thank the referee for this comment. The average height of the first level is approximately 35 meters. Therefore we agree with Mark that considering the emission release within the first level only is not completely unrealistic. We modified the term unrealistic with speculative.

This comment also gave us the idea to do another sensitivity run in which the all the Hg(p) from BB is released in the first layer, whereas the Hg(0) continued to be emitted uniformly in the PBL. Unfortunately, this run did not give any further contribution to the discussion, so we have not included it in the analysis.

5. Figures 6 and 7 are a really interesting way to present the results! However, it took a little time to get my head around what they were saying at first. Perhaps a little more explanation could be added in the caption for these figures?

We added a more detailed explanation to the figures

Agreement maps of high  $\ensuremath{\cells}$  deposition model cells obtained considering only BB emissions and assuming 0\%, 15\% and 30\% to be  $\ensuremath{\cells}$  under both the oxidation mechanisms considered,  $\ensuremath{\cells}$  (a) and  $\ensuremath{\cells}$  (b).

The maps show the areas where deposition is greater than  $\sum u+$

. . .

Agreement maps, under three different speciation scenarios: 0% (a), 15% (b), and 30% (c)  $\eq Hg^{P}$ , of high  $\eq Hg$ } deposition model cells obtained considering only BB and using the  $\eq O_3\Ace{OH}$ , and the  $\eq Br$  oxidation mechanisms, and a sensitivity run where all  $\eq Hg$ } BB emissions were considered inert (i.e. all  $\eq Hg^{P}$ ). The deposition field from for this ``inert`` run was retained under the three different speciation scenarios. The maps show the areas where deposition is greater than  $\medsing Mmu+\sigma$ .

**Technical corrections and/or suggestions**

(...Note that in the following, if a wording change or other correction is being suggested, I have simply included the final wording being suggested, rather than any sort of "track changes" notation. Apologies if this leads to any lack of clarity.)

• P1.L22. "Its relative importance may increase in the coming years, e.g., if the Minimata Convention and/or other efforts lead to reductions in anthropogenic emissions."

**We prefer to maintain the original sentence.**

• P2.L16-17. "...resulting from BB, when variations in HgP fractions and production processes are considered."

**We implemented the suggestion.**

• P2.L17-19. "The most recent version of the GFED BB emission inventory (van der Werfet al., 2010; Randerson et al., 2012; Mu et al., 2011), has been included in the global online Hg chemical transport model ECHMERIT, to simulate Hg deposition from BB for the year 2013 and to quantify the influences of variations in model inputs, assumptions and parameterisations."

**We implemented the suggestion.**

• P2.L23. "... version of the inventory..."

**We corrected it.**

• P2.L24. Need period at end of sentence.

**We corrected it.**

• P2.L27. Wouldn't these be considered "sensitivity" runs, rather than "control" runs?

**We implemented the suggestion.**

• P2.L27. "... see Andela et al. (2013) (and references therein) for a description..."

**We modified the sentence.**

• P3.L4. "Unless explicitly stated,..."

**We removed this sentence.**

• P3.L9. "This value is within the range of observations (Obrist et al., 2007; Finley et al., 2009). However, since there are uncertainties in Hg speciation from BB (Zhang et al., 2013), further simulations were carried out with varying fractions of HgP (0%, 4% and 30%)."

**We implemented the suggestion.**

• P3.L15-17. "The principal vertical profile used (PBL-Profile) maps Hg emissions uniformly within the Planetary Boundary Layer (PBL), whereas in the second, the vertical profile of the standard version of the ECHAM-HAM model was used (HAM-Profile)(Zhang et al., 2012)."

**We implemented the suggestion.**

• P3.L16. Could the "HAM" acronym be defined the first time it's used?

**HAM refers to the complete aerosol module coupled to ECHAM6 in the ECHAM6-HAM model, but it seems to be Hamburg Aerosol Model, it was developed at the MPI-Hamburg, but we can't find this is any of the publications.**

• P3.L21-23. "These simulations primarily employ a O3/OH Hg0(g) oxidation mechanism. However, since the precise atmospheric Hg oxidation mechanism remains unclear (Hynes et al., 2009; Subir et al., 2011, 2012; Gustin and Jaffe, 2010; Gustin etal., 2015), a number of runs were performed using a Br-based oxidation mechanism."

**We implemented the suggestion.**

• P3.L28-29. "Finally two simulations were conducted including Hg emissions from all sources and including re-emissions, to evaluate model performance against measurements (see Appendix A)."

**We changed this section in the revised text.**

• P3.L28-29. What additional emissions were used for these "all-source" simulations?

**We modified the text to explain the Hg sources included:**

• P3.L32. "The majority of Hg releases from BB is believed to occur as Hg0(g)."

**We prefer to maintain the original sentence.**

• P4.L7. "properties" is misspelled.

**We corrected it.**

• P4.L13-16. What equation(s) were used, with what parameters? That is, you say that the Hg0 to HgP ratio is determined by FMC, but what is the mathematical relationship used?

**The partitioning were calculated dynamically using the piece wise linear relationship between Fuel Moisture Content empirically determined from the relative figure in Obrist et al., 2007.**

• P4.L27. I cannot really see very many "notable differences" in Figures 1 and 2. Part of the issue is that the figures are very small and the color ramp does not have a lot of contrast. Could the figures be bigger?

**We thank the referee for this useful feedback. We will upload the images at the maximum resolution allowed. Moreover in the revised text we included a new figure showing the ration between Hg(p) and Hg(0) emissions for all relevant cases where the differences are more evident.**

• P5.L3. At a number of points in the document, it is stated that only the 85:15 emissions speciation results are shown "for clarity". Its not clear to me why showing the results for other speciation profiles would make things less clear. There would be more figures, but would clarity really suffer?

We reported only the 85:15 emissions speciation, since the ratio between two species remains constant over the entire space domain. However we have added a new Figure showing the geographical distribution of the ratio Hg(p):Hg(0) for all relevant cases, and the latitudinal profile in a new panel in the new Figure 4. This allows for a quick comparison for all emissions assumptions considered.

• P5.L14. How were the latitudinal deposition profiles normalized?

**We normalized the latitudinal profiles by the maximum value. We include this detail in the revised text.**

• P5.L18. What is the height of the first model level?

**On average approximately 35 meters. We included this detail.**

- P5.L20-22. A few comments about the following sentence: "This last vertical distribution scenarios are unrealistic, however the differences obtained here contrast with the findings of De Simone et al. (2015) and are due to the fraction of HgP included in this study."
- Not exactly sure what you are trying to say here in terms of comparison to findings of De Simone et al. (2015).
- As noted above in the Specific Comments, I'm not sure I agree that the vertical distribution being referenced is unrealistic.
- This sentence needs to be reworded somewhat for grammar and clarity.

**We finally decided to delete this sentence from the revised text.**

• P5.L28. Do you mean the "deposition peak"?

**We reworded the sentence:**

The emission peak at around 50\$\degree\$N remains relatively distinct also in the deposition for all the simulations

• P6.L1. Maybe would be clearer if the section was called something like this: "Impact of atmospheric oxidation pathway and speciation profiles on geographic distribution of deposition".

**We thank the referee, however we prefer to maintain the original title of the section.**

• P6.L10-12. This sentence is a little confusing, particularly with the use of "all" towards the end. This "all" confused me before I realized you didn't really mean "all".

**We corrected it.**

• "To better understand the combined effect of Hg speciation and oxidation pathway on the deposition distribution, agreement maps were created, to highlight the model cells where different simulations all predict significant deposition..." Maybe better to say something like this: "To better understand the combined effect of Hg speciation and oxidation pathway on the deposition distribution, agreement maps were created, to highlight the similarities and differences in the distribution of high-deposition model cells in different simulations..."

**We implemented the suggestion.**

• P6.L12. What statistical distribution is the "standard deviation" calculated for, e.g., is it the combined data set of cell-by-cell deposition for all cells in all relevant simulations?

**It is exact: it is calculated for all cells in all relevant simulations.**

• P6.L14. "Using the O3/OH mechanism, the number of model cells in which the model predicts high deposition..."

**We implemented the suggestion.**

• P6.L21. maybe "contrasts" (or simply "presents") rather than "confronts"

**We implemented the suggestion.**

• P6.L22. Not sure what you mean by "passive tracer" in this context. It still deposits, right? In other simulations, how are HgP emissions not like a "passive tracer" in the same context? I guess you are implying here that there is no chemical reactions in which Hg0 is oxidized to HgP, and/or that there are no processes converting HgP to another form of Hg. And so, there should be no impact of the oxidation mechanism chosen. But, as noted above in the specific comments above, you could add some additional detail to the text regarding these and other issues to make things clearer.

**We corrected the term passive with the more exact inert.**

• P6.L26. Seems like maybe this section could be divided into two. One called "Uncertainty" and one called "Biomass Burning versus Anthropogenic Impact"

**We thank the referee, however we prefer to maintain the original organization for the section.**

• P7.L1. Could refer the reader to the figure or table that shows the point you are making. Also, instead of "actually have no influence", could say something like "have little influence". And as noted above, you haven't convinced me that the emissions into the first model level – or at least emissions into something less than the full PBL – are really "unrealistic".

**We corrected it.**

• P7.L7. I don't see the Antarctic in the tabular results, but you give results here?

**We refer to the Southern Ocean. We corrected it.**

• P7.L12. "... as in De Simone et al. (2015)." (and same correction a few lines later)

**We corrected it.**

• P7.L13-16. What is an "inspected ensemble"? How was the eventual ensemble created – medians of values for each cell, or mean values for each cell, or some other method?

**An inspected ensemble is an ensemble constructed excluding redundant information, i.e. excluding the runs that give very similar results. The ensemble is created by the mean values for each cell.**

• P7.L24. "just about everywhere" (seems like there are a few locations less than 25%?)

**We agree with the referee, but we want to underline the higher relative contribute in the SH.**

• Table 1. Model Evaluation (not Model Validation)

**We modified the structure of the table**

• Table 2. Would be helpful to explain the "R" and "P-KS" parameters a little either in the Table or in the text. At least to me, it seems a little too cryptic.

**We included the description in the table.**

• Table 3 and Table 4. Maybe could make these into some sort of graphic, either instead of or in addition to?

**We thank the referee, however we prefer to maintain the tables. There are a lot of figure in the text.**

• Table 5. What measurement sites? How many sites? What networks? What averaging time for "r" and for "NMRSE%"? Need some more detail here. What about comparison against HgP measurements? This would seem to be important for this paper!

In the revised text we have included a new subsection within section 3 dedicated to the comparison with Hg measurements from the GMOS network for 2013, to validate the model, and to assess any feedbacks/constraints related to the different assumptions considered about the Hgp emissions from BB. More particularly, when considering the Hg emissions from all other sources, the very small perturbation produced by moving a fraction of Hg BB emissions from Hg0 to Hgp in almost all sensitivity runs causes very little perturbation to the TGM and wet deposition results. Conversely the Hgp in air concentration samples collected in a number of sites from GMOS networks for the year 2013 enabled us to assess the impact of Hgp emissions from BB and to distinguish between the different assumptions. In particularly at two remote sites the model runs including a fraction of Hg(p) from fires resulted in a better agreement with measurements.

We included the new Section 3.4 "Constraints from Global Measurements networks" see page 7 of the revised paper.

• Figure 3, and in fact, most figures: Why so small? For Figure 3, could make it much wider and I think would be much clearer. Difficult to see data when lines overlap so much. Maybe consider some sort of differential dotted/dashed line(s) so that they might be able to be distinguished even when congruent?

We thanks the referee for this useful feedback. See above. We thank the referee for the suggestion, however we believe that using different style for the lines will be more confusing.

Andela, N., Kaiser, J., Heil, A., van Leeuwen, T., van der Werf, G., Wooster, M., Remy, S., and Schultz, M.: Assessment of the Global Fire Assimilation System (GFASv1), MACC-II (Monitoring Atmospheric Composition and Climate) project, 2013; http://juser.fz-juelich.de/record/186645.

Jung, G., Hedgecock, I., and Pirrone, N.: ECHMERIT V1. 0–a new global fully coupled mercury-chemistry and transport model, Geosci. Model Dev., 2, 175–195, 2009.

CAMx: CAMx, user's guide, version 4.40, Environ International Corporation, California, 2006.

Slinn, S. A. and Slinn, W. G. N.: Prediction for particle deposition of natural waters, Atmos. Environ., 14, 1013–1016, 1980.

Seinfeld, C. P. and Pandis, S. N.: Atmospheric Chemistry and Physics, From Air Pollution to Climate Change, 1998.

Janhäll, S., Andreae, M. O., and Pöschl, U.: Biomass burning aerosol emissions from vegetation fires: particle number and mass emission factors and size distributions, Atmos. Chem. Phys., 10, 1427-1439, doi:10.5194/acp-10-1427-2010, 2010.

Amos, H. M., Jacob, D. J., Holmes, C. D., Fisher, J. A., Wang, Q., Yantosca, R. M., Corbitt, E. S., Galarneau, E., Rutter, A. P., Gustin, M. S., Steffen, A., Schauer, J. J., Graydon, J. A., Louis, V. L. S., Talbot, R. W., Edgerton, E. S., Zhang, Y., and Sunderland, E. M.: Gas-particle partitioning of atmospheric Hg(II) and its effect on global mercury deposition, Atmospheric Chemistry and Physics, 12, 591–603, 2012.

Steffen, A., Bottenheim, J., Cole, A., Ebinghaus, R., Lawson, G., and Leaitch, W.: Atmospheric mercury speciation and mercury in snow over time at Alert, Canada, Atmospheric Chemistry and Physics, 14, 2219–2231, 2014.

---

## Author Comment (AC3) · 22 Dec 2016

De Simone et al. (doi:10.5194/acp-2016-685) provide a very detailed model sensitivity study on the influence of partitioning of particulate mercury from biomass burning on its deposition patterns. Such partitioning effect has not been incorporated into most mercury chemical transport models, but it is worthy of attention in the mercury community. The topic of this study is well within the scope of ACP. However, I think the authors should address the following general and specific comments before its consideration of publication.

We thank the referee for his positive general comments and also for his specific comments and feedbacks that helped us to improve the general quality of the manuscript.

General comments:

(1) A major weakness of this manuscript is lack of model observation comparison. The authors point out several significant differences of the deposition fluxes in different model scenarios. Do the available observations provide constraints on the parameterizations of mercury BB emissions?

In the revised text we have included a new subsection within section 3 dedicated to the comparison with Hg measurements from the GMOS network for 2013, to validate the model, and to assess any feedbacks/constraints related to the different assumptions considered about the Hgp emissions from BB. More particularly, when considering the Hg emissions from all other sources, the very small perturbation produced by moving a fraction of Hg BB emissions from Hg0 to Hgp in almost all sensitivity runs causes very little perturbation to the TGM and wet deposition results. Conversely the Hgp in air concentration samples collected in a number of sites from GMOS networks for the year 2013 enabled us to assess the impact of Hgp emissions from BB and to distinguish between the different assumptions. In particularly at two remote sites the model runs including a fraction of Hg(p) from fires resulted in a better agreement with measurements.

We included the new Section 3.4 "Constraints from Global Measurements networks" see page 7 of the revised paper.

(2) It has been suggested that the partitioning of mercury in the atmosphere depends on temperature and aerosol concentrations (for example, Amos et al., 2012). What is the treatment in this study and what is its scientific basis?

In the base configuration of the model Hg(p) is assumed to be inert, it is not considered a product of Hg(0) oxidation. It is emitted from either anthropogenic sources or BB, and it is subject to transport and deposition processes only. However some studies (Steffen et al., 2014, Amos et al, 2012) have suggested that a partitioning of reactive Hg (i.e., Hg(II)) between gas and particle might occur. In particular it has been suggested that this process could be driven by air temperature and availability of aerosol particles (Amos et al, 2012). Therefore, two other simulations were conducted including this temperature dependent gas-particle portioning, to assess the impact of considering a fraction of Hg from BB as Hg(p) under this assumption.

We modified the text opportunely

 $ce{Hg^{P}}$  is assumed to be inert, whenever it is emitted from anthropogenic or BB activities, is subject to transport and deposition processes and it is not involved in any chemical reactions.

•••

Some studies \citep{Steffen2014,Amos2012} suggested that the partitioning of reactive specie between gas and particle might be driven by air temperature and on availability of aerosol particles. Therefore, two other simulations were conducted including the temperature dependent gas particle partioning described by \citet{Amos2012}, one assuming BB \ce{Hg} emissions to be only  $ce{Hg^{0}_{0}}$ , and another assuming a 15\% of BB \ce{Hg} emissions as  $ce{Hg^{P}}$ .

(3) More details of the model parameterizations should be provided. A key process is the photo-reduction of oxidized mercury in the atmosphere. Does the model allow such process in this study? Would this process affect the major conclusion of this study?

**We extended sections 2.2 and 2.3 to describe better the parameterizations included in the model, both in the base configuration or in the variants considered.**

In particular, the atmospheric reduction of Hg reactive species to Hg(0) has been included in different modeling studies, including De Simone et al, 2014, to regulate the atmospheric residence time of elemental Hg and to finally best match the observations. The mechanisms that have been proposed are many, including the photo-reduction of the oxidized Hg. However some of them are unlikely to occur under most atmospheric condition, see Know and Selin 2016 for a recent review. Due to these large uncertainties, we preferred not to include the reduction in this study.

**We included this explanation in the revised paper:**

ECHMERIT, in the base configuration, includes the  $c\{Hg^{0}_{(g)}\}$  oxidation of in  $c\{Hg^{II}_{(g/aq)}\}$  oxidation by  $ce\{O_{3}\}/ce\{OH\}$  in the gas and aqueous phases. OH and  $O_{3}^{s}$  concentration fields were imported from MOZART (Model for Ozone and Related chemical Tracers)  $citep{Emmons2010}$ .

 $ce{Hg^{P}}$  is assumed to be inert, whenever it is emitted from anthropogenic or BB activities, is subject to transport and deposition processes and it is not involved in any chemical reactions. The  $ce{Hg^{P}}$  log-normal particle size distribution is subdivided into a fixed number of size intervals. Details can be found in  $citep{Jung2009}$ . Beyond this standard configuration a number of alternative processes and chemical mechanism has been considered for this study, as explained in  $ref{subsec:sim_and_scopes}$ .

Atmospheric reduction of  $ce{Hg^{II}_{(g/aq)}}$  to  $ce{Hg^{0}_{(g)}}$  has been included in many models to regulate the residence time of  $ce{Hg^{0}_{(g)}}$  in the atmosphere. However, a number of the proposed mechanisms are unlikely to occur under most atmospheric conditions, or are based on empirical rates to better match the observations, see  $citet{Kwon2016}$  for a recent review. Due to this uncertainty, reduction was not included in this study.

Specific comments:

(1) Title: I suggest changing "during" to "from".

**We thank the referee for this suggestion.**

(2) Page 2, line 31: What is the global average enhancement ratio? Does it fit in the observed range (for example, Slemr et al., 2014)?

Biomass Burning emissions of Hg(0), in all cases, are calculated from CO emissions of GFED (or the relevant inventory) by an uniform global enhancement ratio (ER) of 1.96 x 10-7 as given by Fried et. al

**2009, calculated averaging the ERs obtained by measurement for different biome and areas. It is well within the overall observed range, as recently reviewed also by Wang et al., 2015.**

**The revised text reads:**

 $\ensuremath{ce{Hg}}\$  emissions from BB were included in the model by mapping them to CO emissions using the global averaged Enhancement Ratio of \$1.96\times10^{-7}\$ as obtained by \citet{Friedli2009} averaging field measurements from biome and areas globally distributed, including in plume measurements from CARIBIC project \citep{Ebinghaus2007}. Other previous modeling studies included different ERs \citep{DeSimone2015, Holmes2010}, however all these values are well within the uncertainties (\$0.3-6.0\times10^{-7}\$, see \citet{Wang2015}).

(3) Page 3, lines 15-20: I do not quite understand why these two schemes of vertical profiles are equal less than 4 km. Could more explanations be given here?

We thank the referee for pointing out this error within the text. The two schemes actually are equal when the PBL height is greater than 4km. This threshold value is purely arbitrary, but it is the standard configuration in ECHAM6-HAM2.

**We corrected the error in the revised text:**

The HAM-Profile is equal to PBL-Profile when the PBL height is greater than 4000 \$m\$, otherwise 75\% of the emissions are placed within the PBL, and the remainder in the two layers above the PBL (17 and 8\%). This threshold value is arbitrary, however is the standard configuration of ECHAM6-HAM2 \citep{Zhang2012ham,Veira2015}}.

(4) Sect. 2.4: Are there any statistical relationships among OC, PM, and FMC? I am curious since they are all linked to the combustion characteristics.

In the revised text we included a new figure (the figure 3) showing the ratio between Hg(p) and Hg(0) annual BB emissions under the three scenarios PM, OC, and FMC. The distribution of the resulting ratio is different among the scenarios, but they agree on regions where the Hg(p) is relatively the highest, especially for OC and FMC, particularly in the NH. This could be related to the combustion characteristics in those areas where the FMC is the highest, generally yielding lower flame temperatures, smoldering-phase combustion, that in turn yields higher emissions of OC (Zhang et al., 2013).

**In the revised text we add the following discussion:**

Referring now to the panels of the Fig. \ref{fig:RATIO\_EM}, it is evident how the geographical distributions of the ratio of the emissions between  $ce{Hg^{P}}$  and  $ce{Hg^{0}}$  are different among the assumption considered. However for  $ce{OC}$  and FMC they generally agree on areas where the  $ce{Hg^{P}}$  emissions are relatively the greatest, especially in the North Hemisphere, and particularly for areas above \$60\degree\$N. The agreement between  $ce{OC}$  and FMC is not surprising and related to the combustion characteristics that enhance the  $ce{OC}$  emissions, i.e. the lower temperatures and the dominant smoldering phase of combustion  $cite{Zhang2013}$ , that are likely to occur where the FMC is the greatest.

(5) Page 4, line 29: Could more explanations be given about the differences of the emission (and also the deposition) patterns > 60-degree north in difference scenarios (mapping to OC vs FMC)?

As above, the relatively higher Hg(p) emissions in areas > 60 –degree north in both the OC and FMC scenarios are likely to be related to the existing linkage between combustion characteristics in areas with the highest FMC and the processes yielding to an increases of OC emissions.

We extend the discussion in the revised text.

As more evident in Fig.  $ref{fig:Lat\_EM}(c)$ , the most notable differences among the different assumptions hypothesized, are above \$60\degree\$N, where both the \ce{OC} and the FMC cases agree on determining the greatest \ce{Hg^{P}} emissions probably due to the linkage between \ce{OC} emissions and combustion processes favored by FMC \citep{Zhang2013}, and between \$30\degree\$S and \$45\degree\$S, where only \ce{OC} agree, probably due to different processes.

(6) Figure 4: It seems that the influences of different parameterization of PBL-type vertical profiles and different temporal resolutions are insignificant. Could these be due to the gross spatial and temporal resolutions of the model used in this study?

**These two assumption leads to very similar results. We agree with the reviewer that this could be due to the coarse spatial and temporal resolution of the model.**

**We underline this in the revised text:**

Figure \ref{fig:Lat\_DEP}(a) demonstrates the very limited impact of the time resolution used for BB emissions, probably due to the coarse horizontal resolution of the model.

Editorial comments:

(1) Page 1, line 17: add brackets for "Hg". (2) Page 1, line 23:

"asses" should be "assess". (3) Page 2, line 6: add a comma before "however". (4)

Page 2, line 27: wrong reference format. (5) Page 4, line 2: "is of great importance".

(6) Page 4, line 28: "emissions". (7) Page 5, line 11: "where" should be "were". (8)

Page 6, line 28: remove comma. (9) Page 7, line 12, 15: wrong reference format. (10)

Page 8, line 14: remove "the"; full name of "TGM".

**We addressed all editorial comments in the revised text.**

Kwon, S. Y. and Selin, N. E.: Uncertainties in Atmospheric Mercury Modeling for Policy Evaluation, Current Pollution Reports, 2, 103–114, 2016.

Wang, Y., Huang, J., Zananski, T. J., Hopke, P. K., and Holsen, T. M.: Impacts of the Canadian forest fires on atmospheric mercury and carbonaceous particles in northern New York, Environ. Sci. Technol., 44, 8435–8440, 2010.

Zhang, Y., Obrist, D., Zielinska, B., and Gertler, A.: Particulate emissions from different types of biomass burning, Atmos. Environ., 72, 27–35, 2013.

---

## Author Comment (AC1)

Review of "Particulate-Phase Mercury Emissions during Biomass Burning and Impact on Resulting Deposition: a Modelling Assessment" by Francesco de Simone.

De Simone and co-authors have explored the sensitivity of an atmospheric mercury model (ECHMERIT) to assumptions about mercury emissions from biomass burning. The main focus of their sensitivity tests is the fraction of mercury that is emitted as Hg(p) vs. Hg(0), although they also test model sensitivity to emission time resolution and oxidants for Hg(0). They use several different plausible Hg(p) fractions (0 to 30%) and various way to apportion that fraction (constant or proportional to biomass burning CO, PM, or OC). The partitioning of emissions is an important issue, as the authors explain, because Hg(0) has a long atmospheric residence time and circulates globally while Hg(p) has a short residence time and deposits near the emission source. These are reasonable sensitivity tests and I expect that other mercury scientists and modelers will be interested in the results.

**1)** The main weakness of the paper is that it provides no comparison to observations, except for an unexplained and unused table in the Appendix. The paper therefore provides no insight into which, if any, of the many model configurations provide reasonable comparisons with observations. There are abundant surface and aircraft measurements of Hg(0), Hg(p), CO, OC, and PM that could be used for this purpose (GMOS, AMNET, CARIBIC, ARCTAS, INTEX-B). If the ECHMERIT model is run in a climate mode, so that it does not match the daily weather conditions at measurement sites, the simulated distributions and correlations between multiple species can still be compared with observations. Without comparison between model and observations, I do not think that this paper in its current form is suitable for publication in ACP.

**We thank the referee for his/her positive general comments and also for his/her specific comments and feedbacks that helped us to improve the general quality of the manuscript.**

In the revised text we have included a new subsection within section 3 dedicated to the comparison with Hg measurements from the GMOS network for 2013, to validate the model, and to assess any feedbacks/constraints related to the different assumptions considered about the Hg(p) emissions from BB. More particularly, when considering the Hg emissions from all other sources, the very small perturbation produced by moving a fraction of Hg BB emissions from Hg(0) to Hg(p) in almost all sensitivity runs causes very little perturbation to the TGM and wet deposition results. Conversely the Hg(p) in air concentration samples collected in a number of sites from GMOS networks for the year 2013 enabled us to assess the impact of Hg(p)emissions from BB and to distinguish between the different assumptions. In particularly at two remote sites the model runs including a fraction of Hg(p) from fires resulted in a better agreement with measurements.

We included the new Section 3.4 "Constraints from Global Measurements networks" see page 7 of the revised paper.

**2)** Another significant problem with the current version of the manuscript is that the methods do not contain enough detail to understand how the emissions were constructed. How are biomass burning emissions of Hg (=Hg(p)+Hg(0)) calculated from the biomass burning CO or DM provided by GFED? Please provide the relevant emission factors or emission ratios. In simulations where Hg(p) fraction depends on OC, PM, or FMC the manuscript needs to clearly explain how the Hg(p) fraction is calculated from OC, PM, or FMC. Do the emission factors (e.g. CO/DM, OC/DM) vary geographically with biome type? A simulation with 100% Hg(p) from biomass burning is discussed in Sect. 3.3 but not described in the methods. Regardless of

how Hg is calculated in the emission inventory, please report the Hg/CO ratio because this would enable comparison to many observations that are reported this way.

**Biomass Burning emissions of Hg(0), in all cases, are calculated from CO emissions of GFED (or the relevant inventory) by an uniform global enhancement ratio (ER) of 1.96 x 10-7 as given by Friedli et. al 2009, calculated averaging the ERs obtained by measurement for different biome and areas.**

**The text has been modified appropriately in the revised manuscript.**

 $\label{eq:generalized_states} $$ \eqref{Hg} emissions from BB were included in the model by mapping them to CO emissions using the global averaged Enhancement Ratio of $1.96\times10^{-7}$ as obtained by \citet{Friedli2009} averaging field measurements from biome and areas globally distributed, including in plume measurements from CARIBIC project \citep{Ebinghaus2007}. Other previous modeling studies included different ERs \citep{DeSimone2015, Holmes2010}, however all these values are well within the uncertainties ($0.3-6.0\times10^{-7}$, see \citet{Wang2015}).$

The Hg(p) emissions are calculated from CO, OC and PM GFED emissions, based on the respective scenario investigated. We have added a new appendix to describe in detail the methods used to calculate the different emission fields used in this study.

**In the revised text we modified the relevant sections to clarify all these details.**

"The ways how the different \ce{Hg} BB emission fields are calculated are detailed in the Appendix \ref{app:B}."

\section{How Hg emission fields are calculated} %% Appendix A

**\subsection{Mapping to CO}**

When mapped to  $ce{CO}$ , the emissions of  $ce{Hg^{0}}$  were calculated from those of  $ce{CO}$  using a global averaged ER (\$1.96\times10^{-7} mol/mol\$). These were unchanged in the run assuming  $ce{Hg}$  emissions from BB to be \$100\%\$  $ce{Hg^{0}}$ , whereas were opportunely fractioned between  $ce{Hg^{0}}$  and  $ce{Hg^{P}}$  species to be in the ratio \$96:4\$, \$85:15\$, and \$70:30\$, in mass, in the runs considering the respective constant fractions of  $ce{Hg^{P}}$ . Consequently, the geographical and temporal distributions of  $ce{Hg^{0}}$  and  $ce{Hg^{P}}$  BB emissions follow those of  $ce{CO}$ . For all cases, the GFEDv4 inventory was used based, except for those sensitivity runs performed to test the impact of different inventories (i.e. the FINNv1.5 and the GFAS1.4), which used the respective inventories.

\subsection{Mapping to OC}

When mapped to  $ce{OC}$ , geographical and temporal distributions of  $ce{Hg^{0}}$  BB emissions, as well as the total  $ce{Hg}$  emitted, were calculated in the same way as described in Appendix  $ref{app:subMCO}$ . The fractioning of  $ce{Hg}$  emissions, in mass, between  $ce{Hg^{0}}$  and  $ce{Hg^{P}}$  species were assumed to be in the ratio \$85:15\$. The  $ce{Hg^{P}}$  emissions so calculated were then geographically and temporally mapped to those of  $ce{OC}$  from GFEDv4 inventory.

\subsection{Mapping to PM}

This mapping method is similar to one described in Appendix \ref{app:subMOC}, except for the fact the  $\eq Hg^{P}$  temporal and geographical distributions follow those of  $\eq PM$  from GFEDv4 inventory.

**\subsection{Emissions speciation determination by FMC}**

When using this procedure for determining the BB emissions speciation between  $c{Hg^{0}}$  and  $c{Hg^{P}}$  species, the geographical and temporal distributions of  $c{Hg^{0}}$  and  $c{Hg^{P}}$  BB emissions, as well as the total  $c{Hg}$  emitted, were calculated in the same way as described in Appendix  $ref{app:subMCO}$ . The main difference is in that the fractioning of  $c{Hg}$  emissions, in mass, between  $c{Hg^{0}}$  and  $c{Hg^{P}}$  species were calculated dynamically using the piece wise linear relationship between Fuel Moisture Content empirically determined by relative figure in  $c{t}{O}$ .

As a proxy for FMC, we used the monthly averaged vegetation water content (VWC) derived from passive microwave remote sensing data (Advanced Microwave Scanning Radiometer 2 (ASMR2)), and employing the Land Parameter Retrieval Model (LPRM) available at (\url{http://gcmd.nasa.gov/search/Metadata.do?Entry=C1235316240-GES\_DISC\#metadata}).

**Regarding the CO/DM, OC/DM and PM/DM emissions factors, we use the biome based EF provided with the GFED4 script based on Akagy et al., 2011, as partially explained in Section 2.2.**

**We have modified the text:**

A script is provided to derive gaseous and particle emissions from DM fields making use of bioma based emissions factors based on \citet{Akagi2011} and \citet{vanderWerf2010}.

Other issues:

3) Is there chemical reduction in the model? If so, is Hg(p) affected by it?

Atmospheric reduction of Hg reactive species to Hg(0) has been included in different modeling studies, including De Simone et al, 2014, to regulate the atmospheric residence time of elemental Hg and to optimise the comparison with observations. However a number of reduction mechanisms have been proposed and some of them are unlikely to occur under most atmospheric conditions. Due to this uncertainty, we preferred to not include the reduction in this study.

**We included this explanation in the revised paper:**

Atmospheric reduction of  $ce{Hg^{II}_{(g/aq)}}$  to  $ce{Hg^{0}_{(g)}}$  has been included in many models to regulate the residence time of  $ce{Hg^{0}_{(g)}}$  in the atmosphere. However, a number of the proposed mechanisms are unlikely to occur under most atmospheric conditions, or are based on empirical rates to better match the observations, see  $citet{Kwon2016}$  for a recent review. Due to this uncertainty, reduction was not included in this study.

4) The total Hg emissions from biomass burning in this work are 400 Mg/yr. A previous analysis by the same authors reported much higher mean emissions of 675 Mg/yr (Pirrone et al., 2010). What is the reason for such a large change?

Naturally, Biomass Burning activity and associated emissions are subject to a strong year to year variability. More particularly the methods by which the activity retrieved is analyzed and the emissions of the different chemicals are estimated are subject to large uncertainties regarding both the DM and Carbon emissions and EFs being used. Moreover these are often revised due to new field measurements available and to technical advances in retrieval algorithms.

Just to give some details, looking at the historical GFED4 yearly estimated DM and C emissions, the ratio between maximum and minimum over the period 1997-2015 is about 1.7. For gases and particles, the BB emissions also depends on (the revision of) EFs used, so the differences can be greater. For example, regarding the CO GFED4 BB emissions estimates, from which Hg emissions are calculated, this ratio over the same period is greater than a factor 2.

Regarding the comparison with the annual averaged Hg emissions from BB reported in Pirrone et al 2010, it refers to the estimation calculated from version 2 of the GFED (Friedly et al. 2009). As reported in Van der werf et al., 2010, yearly estimated CO emissions of the revision 3 of GFED were found to be lower on average by 13%, and in some years by more than 50%.

We modified opportunely the text to explain this difference:

The total \ce{Hg} emitted in 2013 based on the GFED inventory is roughly \$400\$\,Mg, at the lowest end of the initial estimates (\$675 \pm 240\$\,Mg) \citep{Friedli2009}, but reasonable considering the natural variation of BB activity and the trend in diminishing the \ce{CO} emissions estimates of the latest inventory revisions (up to 50\% for some years) \citep{vanderWerf2010}.

5) None of the figures show the spatial map of Hg(p)/Hg ratio (or Hg(p)/Hg(0) ratio), which is the central focus of the paper. In addition, all 4 panels of Figure 2 are visually indistinguishable (and indistinguishable from Fig 1, except for magnitude). I think this space would be better used to show the Hg(p)/Hg emission ratios under the various schemes based on CO, PM, and OC.

**In the revised text we have added a new figure (Figure 3) showing the ratio between Hg(p) and Hg(0) emissions for all relevant cases. Moreover we added panel (c) in the new Figure 4 to show the latitudinal distribution of this ratio for all relevant cases.**

Compared to the cases where  $ce{Hg^{P}}$  emissions are mapped to  $ce{CO}$  and  $ce{PM}$  (Figs.  $ref{fig:EM_RM}(a-b)$  and (e-f)), mapping  $ce{Hg^{P}}$  to  $ce{OC}$  and using the FMC to determine the speciation (Figs.  $ref{fig:EM_RM}(c-d)$  and (g-h)) result in enhanced  $ce{Hg^{P}}$  emissions, above \$60\degree\$N, and over some areas of Amazonia, Central Africa and East Asia as is evident in Fig.  $ref{fig:RATIO_EM}$ , potentially impacting the timing and location of deposition to these areas, particularly to the Arctic.

6) Some additional observational studies of Hg in biomass burning plumes should be discussed: Ebinghaus et al., 2007; Holmes et al., 2010.

The results of in plume measurements collected during the CARIBIC aircraft experiment reported in Ebinghaus et al., 2007 are included in of Friedly et al 2009. The ER of  $1.0 \times 10^{-7}$  included in the modeling study of Holmes et al., 2010 is based on limited aircraft measurements in a specific region and is not representing of the biome characteristics at a global scale.

**However we quickly report these reference for the completeness of the review.**

 $\label{eq:hardenergy} $$ \eqref{Hg} emissions from BB were included in the model by mapping them to CO emissions using the global averaged Enhancement Ratio of $1.96\times10^{-7}$ as obtained by \citet{Friedli2009} averaging field measurements from different biomes and regions, including in plume measurements from the CARIBIC project \citep{Ebinghaus2007}. Other previous modeling studies included different ERs \citep{DeSimone2015, Holmes2010}, however all these values are well within the estimated uncertainty ($0.3-6.0\times10^{-7}$, see \citet{Wang2015}).$

**Minor**

• Page 1 Line 3 (P1L3): Add that the Hg which is not Hg(p) is assumed to be Hg(0).

**We have rewritten the sentence to be more clear.**

The greatest fraction of  $ce{Hg}$  from BB is released in the form of elemental Hg ( $ce{Hg^{0}}$ ). However, little is known about the fraction of  $ce{Hg}$  bound to particulate matter ( $ce{Hg^{0}}$ ) released from BB

**• P1L13: 71% to 62% of what?**

**We have rewritten the sentence to be more clear.**

This reduces the fraction of \ce{Hg} from BB which deposits to the world's oceans from 71\% to 62\%.

• P1L15: Statement about mercury in water-stressed and warming forests is speculation that is not supported in the paper.

**We have rewritten the sentence to be more clear**

Under the on-going climatic changes this effect could potentially be exacerbated in the future.

• P1L19: Statement exaggerates the magnitude of biomass burning emissions relative to other anthropogenic emissions; it is certainly less than 1/2 of anthropogenic Hg emissions. First, it is widely acknowledged that a very large portion of biomass burning is anthropogenic, even though emission inventories are not labelled this way. Second, the Muntean et al., 2014 paper does not include mercury emissions from small-scale gold mining, so anthropogenic emissions are much larger than they estimated.

We agree with the referee regarding the anthropogenic to Biomass Burning emission ratio. However Hg emissions from wildfires are not included in anthropogenic emission inventories. More particularly, the reported ratio of the comparison regards the gridded inventories. However we agree to modify the statement to be more conservative.

Although the \ce{Hg} released by BB varies from year to year, it can amount to up to roughly one third of the anthropogenic emission estimates \citep{AMAP/UNEP2013,Friedli2009,DeSimone2015}

Conversely we don't agree with the referee about ASGM, since EDGARv4 contains Hg emissions from Artisanal and small scale gold mining, in fact his is stated in the Abstract of Muntean et al., 2014.

• P3L2: particle emissions are presumably also calculated.

**Corrected.**

• P4L2: "of" great importance

**Corrected.**

• P4L9: Define FMC

**It is defined at its first appearance in Section 1: Introduction**

• P4L23: Is the total Hg emission the same in all simulations? How is it calculated from the GFED DM or CO?

We modified the section to be clearer and we modified the Table 1 to include the total Hg emissions from BB for each run. See above for details.

The exact amount of \ce{Hg} emitted by BB injected in the model for the different runs is detailed in Table \ref{tab:simulations}

• P4L23. "Considering" should begin a new sentence.

**Corrected.**

• P5L14: How are the data in the figures normalized?

**Data are normalized by division by the maximum value. We modified opportunely the caption of the respective figure to explain this..**

• P5L16: It seems very unlikely that the fairly smooth zonal-mean distribution would be altered by finer spatial resolution.

**We have rewritten the sentence.**

Figure \ref{fig:Lat\_DEP}(a) demonstrates the very limited impact of the time resolution used for BB emissions, probably due to the coarse horizontal resolution of the model.

• P6L23: What does "passive tracer" mean? In atmospheric modeling, "passive" usually means that a tracer does not alter the model's transport or physics. (i.e. it is passively transported.) I would therefore expect that all of the Hg species in all of the simulations are passive in this sense.

**We used the term passive tracer to indicate a tracer that is not involved in any chemical transformation. In the revised paper we use the term inert to better describe this property.**

• P6L29: Not quite correct. Oxidant choice still has a big effect on the deposition pattern.

**We reworded the sentence to be more clear.**

Some of the model assumptions and parametrisations, in particular regarding emissions injection into the model layers, made little difference to the eventual deposition fields in the case where emissions from BB were considered to be  $100\%\ce{Hg}(0)_{(g)}\citep{DeSimone2015}.$

• P7L1: Statement says that vertical profile of emissions doesn't matter, but I expect that the vertical profile would be quite important for scenarios with high Hg(p) emission fraction. Like other aerosols and reactive gases, Hg(p) emitted into the free troposphere should disperse much farther than Hg(p) emitted into the boundary layer.

**This is actually due to the small differences between the two main height distributions used. Differences are evident for the sensitivity runs using other height injection assumptions. However these are a little speculative, so we don't include most of them in the final analysis.**

**We reworded the sentences to be more clear.**

However the choice of the two main vertical profile of the BB emissions used for this study, also when combined with the temporal resolution of the emissions actually have no influence on the final Hg total deposition fields, probably due to the limited differences between them. Other cases of emitting all of the emissions into a single model layer do have an impact. However these are a little speculative, and therefore they are not included in the final analysis.

• P7L6: 66% of what?

**It refers to the Hg deposited. We fixed it.**

• Fig 7c: Panel title says "Hg(p) fraction =30%" but one of the plotted quantities is "100% Hg(p)". Only one can be correct.

**We modified both the caption and legend to be more clear.**

• Table 2: How are the correlations calculated? Are they the spatial correlation of the annual mean? Is temporal variability considered in the correlations? What does "Ensemble" mean here?

**We modified the caption to be more clear.**

• Table 3: Title should say "from biomass burning"

**We corrected the title.**

• Table 4: Title says "Mercury deposition (Mg)" but only some rows have units of Mg.

**We fixed it.**

• Table 5: Terms "BASE Full" and "Br Full" are not defined.

**We modified the nomenclature of runs to be more clear.**

van der Werf, G. R., Randerson, J. T., Giglio, L., Collatz, G. J., Mu, M., Kasibhatla, P. S., Morton, D. C., DeFries, R. S., Jin, Y., and van Leeuwen, T. T.: Global fire emissions and the contribution of deforestation, savanna, forest, agricultural, and peat fires (1997–2009), Atmos. Chem. Phys., 10, 11707-11735, doi:10.5194/acp-10-11707-2010, 2010.

Wang, X., Zhang, H., Lin, C.-J., Fu, X., Zhang, Y., and Feng, X.: Transboundary transport and deposition of Hg emission from springtime biomass burning in the Indo-China Peninsula, J. Geophys. Res.: Atmospheres, 120, 9758–9771, 2015.

Akagi, S. K., Yokelson, R. J., Wiedinmyer, C., Alvarado, M. J., Reid, J. S., Karl, T., Crounse, J. D., and Wennberg, P. O.: Emission factors for open and domestic biomass burning for use in atmospheric models, Atmos. Chem. Phys., 11, 4039–4072, 2011.

---

## Author Comment (AC4)

T. Dvonch (Referee)

dvonch@umich.edu

Received and published: 22 September 2016

Review of "Particulate-Phase Mercury Emissions during Biomass Burning and Impact on Resulting Deposition: a Modelling Assessment" by Francisco De Simone et al.

**General Comments:**

De Simone et al. present a detailed assessment of the impact of mercury emissions from biomass burning on resulting atmospheric mercury deposition. The assessment is completed through utilization of an updated emissions database and an updated global mercury chemical transport model. Within this framework, the authors investigate a variety of model parameterizations and the role of other uncertainties on resulting magnitudes and spatial distributions of mercury deposition. Overall, this work represents a sizable effort and further informs the scientific community regarding speciation of mercury emissions, chemical oxidation mechanisms, and spatial and temporal variations, all within the specific context of biomass burning and with relevance to the implementation of mercury policy.

This manuscript represents a substantial contribution to the field, and is very much with the scope of ACP. However, there are several items that could be addressed by the authors and incorporated into revisions that would likely strengthen the manuscript overall.

**We thanks the referee for his positive general comments and for his feedback that helped us to improve the general quality of the manuscript.**

**Specific Comments:**

1. Much of the manuscript focuses on the potential impact of Hg P emissions. However, the issue of particle size is not discussed in the paper. Certainly, there have been assumptions made within the model regarding particle size, with direct implications to the potential transport distance prior to Hg P removal from the atmosphere (via either dry or wet processes). The manuscript would benefit from added discussion specific to particle size.

As discussed in the paper, although there is of experimental evidence that some significant fraction of Hg emitted from BB is bound to particulate, until now in models, Hg emissions from BB were considered only as Hg(0). This is the first modeling study that considers a fraction of Hg from BB to be bound to particulate. Within this scope, the main objective of the paper is to investigate how such speciation impacts the fate of Hg. We consider this study a substantial advance in Hg modeling scientific literature.

Undoubtedly, the particle size distribution will have an impact on the final fate of Hg(p) emitted. However there are large uncertainties regarding the size distribution of particles emitted, and how it evolves during the different phases of BB (see Janhäll and Pöschl, 2010 and the reference therein). Moreover in this first study we prefer to focus mostly on the mechanism related to the emission speciation and on the uncertainty related to some of the processes Hg(p) undergoes in the atmosphere, such as temperature dependent gas-particle partitioning.

*In the revised paper we included the following text:*

No further  $ce{Hg^{p}}$  particle dimension distributions other than the standard log-normal particle size distribution, as described in detail in  $citep{Jung2009}$ , were considered in this study due to large uncertainties regarding the dynamic size range of particle emitted during BB, see  $citet{Janhall2010}$  and the references therein.

2. Table 5 presents summary statistics regarding comparison of model output with available observations from measurement networks. However, there is little text in the body of the manuscript in support of the inclusion of Table 5. The manuscript would benefit from added discussion to characterize and specify how well the model performed in comparison to observations.

In the revised text we have included a new subsection within section 3 dedicated to the comparison with Hg measurements from the GMOS network for 2013, to validate the model, and to assess any feedbacks/constraints related to the different assumptions considered about the Hgp emissions from BB. More particularly, when considering the Hg emissions from all other sources, the very small perturbation produced by moving a fraction of Hg BB emissions from Hg0 to Hgp in almost all sensitivity runs causes very little perturbation to the TGM and wet deposition results. Conversely the Hgp in air concentration samples collected in a number of sites from GMOS networks for the year 2013 enabled us to assess the impact of Hgp emissions from BB and to distinguish between the different assumptions. In particularly at two remote sites the model runs including a fraction of Hg(p) from fires resulted in a better agreement with measurements.

We included the new Section 3.4 "Constraints from Global Measurements networks" see page 7 of the revised paper.

3. No explanation or justification is provided on the selection of 2013 as the model time period. This rationale should be provided in the revisions, along with some indication of the representativeness of 2013 compared to other recent years.

The authors have already investigated many uncertainties related to Hg emissions from BB in De Simone et al., 2015, including the year-to-year Hg BB emission variability for a decade. As explained above, this study focusses on the speciation of Hg emissions from BB, and on the effects on the resulting deposition, and it is investigated for the first time in a CTM. Results for other years could be somewhat different. However we decided to choose 2013 because it was one of the years best covered by measurements within the GMOS project. This allows us to have feedbacks from the comparison with measurements collected at a global scale.

**We modified the text to include the rationale for the choice:**

This study cover a single year, the 2013, which has been chosen due the large availability of measurements from GMOS network \citep{Sprovieri2016\_conc, Sprovieri2016\_wet,Damore2015}.

These results apply for the investigated year (2013) and could be to some extent different considering other years, due to the complex interaction of the numerous actors determining the final fate of \ce{Hg}. However few alternatives of analysis period exist due the limited time coverage of global measurement network(s).

4. Figures 1, 2, 5, & 9 seem to be very instructive. However, they are not easily legible. The size/resolution of these figures should be improved for the benefit of the reader.

We thanks the referee for this useful feedback. We will enlarge the figure at the maximum allowed resolution.

Technical/Editorial Comments:

The units reported in Table 3 need additional clarification (Mg and %).

Page 1 - line 13, "71% to 62%"...of total deposition? Seems this sentence is missing

some needed context.

Page 2 - line 7, should be "fraction of Hg emitted" (add "of").

Page 2 – line 24, period needed after "(Randerson et al., 2012)".

Page 3 – line 8, "equal to the 15%" (remove "the").

Page 4 – line 2, "Hg emissions is of great importance" (add "of").

Page 5 – line 18, "first model level level leads to" (remove "level").

Page 5 – line 19, "approx" should be "approximately".

Page 7 – line 1, instead of "have no influence", perhaps "have little influence"?

Page 8 – line 14, "between the the measurement" (remove "the").

**In the revised paper we have fixed the editorial issues identified by the referee.**

Janhäll, S., Andreae, M. O., and Pöschl, U.: Biomass burning aerosol emissions from vegetation fires: particle number and mass emission factors and size distributions, Atmos. Chem. Phys., 10, 1427-1439, doi:10.5194/acp-10-1427-2010, 2010.